# SP-Mind: An Autonomous Reasoning Agent for Spatial Proteomics Analysis

Yucheng Yuan [1] [*]   Yuanfeng Ji [2] [*]   Zhongxiao Li [2]   Ruijiang Li [2]

## Abstract

Spatial proteomics enables single-cell-resolution characterization of protein expression within tissue architecture, playing a critical role in understanding tumor microenvironments and guiding precision medicine. However, current analysis workflows remain fragmented, requiring expert manual orchestration of heterogeneous tools and limiting research scalability and reproducibility. We present SP-Mind, the first autonomous AI agent designed to unify the spatial proteomics analysis pipeline, from raw multiplexed tissue imaging to downstream phenotype discovery. Equipped with expert-curated biological analysis skills and specialized computational tools, SP-Mind converts natural-language queries into end-to-end analytical workflows without task-specific fine-tuning. To rigorously evaluate its capabilities, we introduce SP-Bench, a comprehensive benchmark spanning diverse tissue types, comprising 102 tasks across 18 distinct categories. Through extensive evaluation on SP-Bench and established downstream tasks, SP-Mind achieves state-of-the-art performance compared to existing open-source biomedical agent baselines.

## 1. Introduction

Recent advances in multiplexed tissue imaging enable the simultaneous detection of dozens of proteins at single-cell resolution (Goltsev et al., 2018; Giesen et al., 2014; Lin et al., 2018). This reveals the "molecular sociology" of tissues: how diverse cell types spatially organize, interact, and collectively shape disease outcomes (Bodenmiller, 2024; Bussi & Keren, 2024). Such insights are critical for precision oncology and immunotherapy (Quail & Walsh, 2024).

Extracting such insights requires a multi-stage computational pipeline: image registration, artifact correction, cell segmentation, marker quantification, phenotyping, and spatial analysis (Bussi & Keren, 2024). Manually orchestrating these stages is prone to errors and impractical at scale. Recent computational pipelines have integrated platform-specific tools (e.g., for CODEX, IMC) into end-to-end workflows, enabling reproducible batch processing (Schapiro et al., 2022; Magness et al., 2024; Buckup et al., 2025). However, two fundamental limitations remain (Bussi & Keren, 2024): (a) Expert-dependent configuration: users must manually select tools and tune parameters based on imaging platform and tissue context; (b) Static execution: these systems cannot dynamically adapt to diverse analytical queries. Given these challenges, a pivotal question arises: *Can we build an autonomous agent that understands user intent, selects appropriate tools and configurations, and executes end-to-end spatial proteomics analysis?*

In this paper, we introduce SP-Mind to address this question. SP-Mind is an autonomous reasoning agent equipped with two core components: (1) a modular library of 10+ specialized tools covering the full spatial proteomics pipeline, from image preprocessing to spatial analysis; (2) expert-curated skill templates that encode domain knowledge for tool selection and parameter configuration. Beyond these, SP-Mind employs a ReAct-style reasoning loop that iteratively observes execution outputs, diagnoses failures, and self-corrects. A single natural-language query triggers SP-Mind to automatically chain tools, complete the analysis, and deliver interpretable results.

To rigorously evaluate agentic spatial proteomics analysis, we introduce SP-Bench, a comprehensive benchmark comprising 102 tasks across 18 categories and 4 difficulty tiers. On SP-Bench, SP-Mind achieves 68.9% execution accuracy, outperforming the strongest baseline by 13 percentage points. We further validate SP-Mind on downstream tasks including cell quantification and annotation, demonstrating consistent improvements over existing approaches.

Our contributions are summarized as follows:

- We systematically define and formalize the problem of automated spatial proteomics analysis for the first time, and we present SP-Mind, the first autonomous

[*]Equal contribution [1]Department of Computer Science, Stanford University, Stanford, USA [2]Department of Radiation Oncology, Stanford University, Stanford, USA. Correspondence to: Ruijiang Li <rli2@stanford.edu>.

*Proceedings of the 43rd International Conference on Machine Learning*, Seoul, South Korea. PMLR 306, 2026. Copyright 2026 by the author(s).

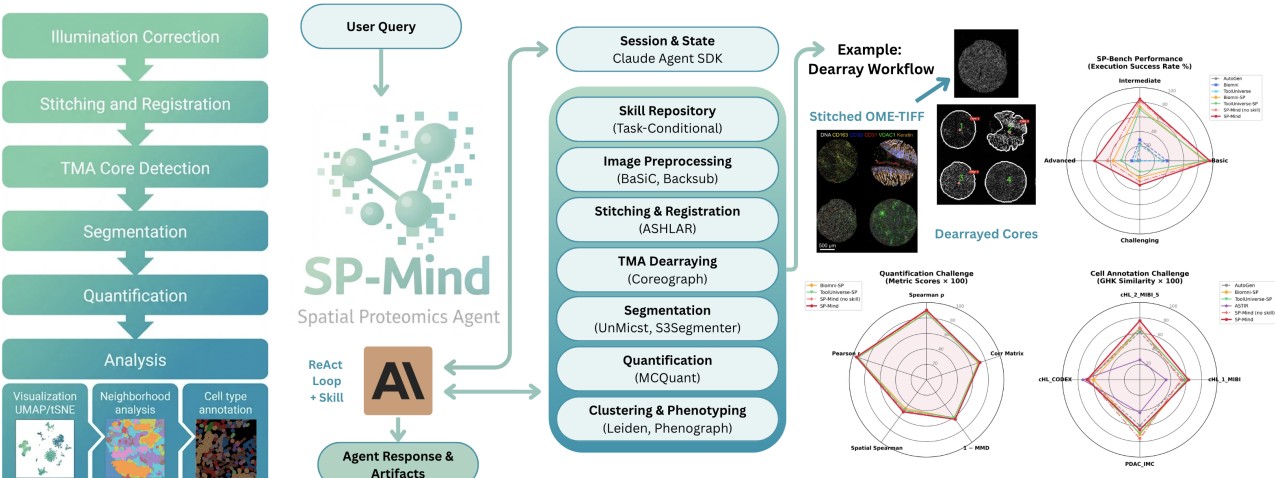

*Figure 1.* **The SP-Mind Framework and Performance Evaluation.** A brief illustration of spatial proteomics workflow and SP-Mind architecture (Left and Center). Benchmarking results on SP-Bench, fine-grained quantification, and cell annotation demonstrate SP-Mind's state-of-the-art performance compared to baseline agents (Right).

AI agent for end-to-end spatial proteomics analysis, equipped with specialized tools and expert-curated skill templates, to address this problem.

- We introduce SP-Bench, the first comprehensive standardized benchmark for evaluating agentic spatial proteomics workflows.

- We demonstrate state-of-the-art performance on SP-Bench and downstream tasks, with extensive ablations validating the effectiveness of our design choices.

## 2. Related Work

**Spatial Proteomics Workflows** Spatial proteomics has tremendous medical potential. However, this translation of raw multiplexed imaging data into biological insight requires a complex, multi-stage computational workflow involving signal calibration (Peng et al., 2017), high-precision image registration (Muhlich et al., 2022), cell segmentation (Yapp et al., 2022), and phenotypic quantification/classification (Bussi & Keren, 2024). To manage this complexity, the field has initiated automation approaches that encapsulate individual processing stages into containerized environments (Boettiger, 2015) orchestrated by workflow management systems like Nextflow (Di Tommaso et al., 2017). MCMI-CRO (Schapiro et al., 2022), TRACERx-PHLEX (Magness et al., 2024), and MARQO (Buckup et al., 2025) exemplify this paradigm, providing an end-to-end pipeline that chains established spatial proteomics and downstream analysis tools. While these frameworks excel at throughput and reproducibility, they suffer from inherent procedural rigidity, lacking the "adaptive intelligence" required to handle the heterogeneity of biological tissues and real-world analysis demands Bussi & Keren (2024).

**Biomedical AI Agents** The emergence of AI agents built upon Large Language Models (LLMs) has fundamentally transformed autonomous reasoning, planning, and tool utilization. Foundational frameworks such as ReAct (Yao et al., 2022), CodeAct (Wang et al., 2024b), and AutoGen (Wu et al., 2024), alongside recent surveys (Xi et al., 2025; Wang et al., 2024a), have codified a standardized agentic architecture comprising three core components: (1) data-oriented observation, (2) LLM-driven reasoning, and (3) code-based action execution. In the biomedical domain, this paradigm has been adapted to create specialized "AI Scientists." (Gao et al., 2024; Qi et al., 2026) Works such as SciAgents (Ghafarollahi & Buehler, 2025), BioDiscoveryAgent (Roohani et al., 2024), CRISPR-GPT (Qu et al., 2026), and GeneGPT (Jin et al., 2024) exemplify this by automating highly domain-specific workflows. More recent approaches have pursued a broader scope, with Biomni (Huang et al., 2025) and ToolUniverse (Gao et al., 2025) pioneering general-purpose biomedical AI assistants. However, despite these advances, a critical gap remains: no existing biomedical agent is capable of orchestrating the end-to-end spatial proteomics analysis pipeline.

**Agent Benchmarks** The rapid proliferation of autonomous agents has necessitated rigorous evaluation frameworks to measure reasoning and tool-use capabilities. In the general domain, benchmarks such as SWE-Bench (Jimenez et al., 2023), GAIA (Mialon et al., 2023), AgentBench (Liu et al., 2023) MMAU (Sakshi et al., 2024), and ToolBench (Qin et al., 2023) have established the gold standard for coding, assessing multi-step planning, and generic API interaction. Parallel efforts have emerged within the biomedical sciences to evaluate domain-specific competence, exemplified by SciBench (Wang et al., 2023), LAB-Bench (Laurent

et al., 2024), BixBench (Mitchener et al., 2025) for scientific problem solving and MedAgentBench (Jiang et al., 2025) for clinical decision-making. However, existing benchmarks exhibit a significant blind spot regarding complex spatial biology workflows. To date, no benchmark exists that targets the holistic spatial proteomics analysis workflow.

## 3. SP-Mind Agent

We present SP-Mind, an LLM-driven reasoning agent for end-to-end spatial proteomics analysis. Given a user query describing an analytical objective (such as cell segmentation, marker quantification, or full pipeline), SP-Mind autonomously decomposes the task into executable steps, invokes appropriate computational tools, and synthesizes results into interpretable outputs. The framework is designed to handle the full complexity of modern spatial biology workflows, which typically require chaining multiple specialized operations with careful attention to intermediate data formats. SP-Mind operates within a sandboxed execution environment with configurable permission levels, supporting both interactive exploration and fully autonomous batch processing.

### 3.1. Agent Architecture and Reasoning

SP-Mind implements a ReAct-style reasoning loop (Yao et al., 2022), iterating through cycles of (1) observation, (2) thought, and (3) execution, until task completion (the agent generates a textual/visual summary for user). In the observation phase, the agent ingests both user-provided task specifications and execution outputs from prior actions, grounding subsequent reasoning in concrete computational state; the agent architecture provides automatic context compaction to manage long-horizon interactions. Following a data-first reasoning protocol, the agent prioritizes inspecting data structures and distributions before committing to analytical actions, reducing cascading errors from misspecified thresholds. In the thought phase, the agent performs explicit chain-of-thought reasoning to interpret observations, diagnose failures, and plan next steps; this deliberative process is externalized through dedicated reasoning blocks, enabling interpretable decision traces for complex multi-step workflows. In the execution phase, the agent takes action through a hybrid code-execution strategy: rather than invoking tools solely through predefined API schemas, SP-Mind uses CodeAct-style paradigm (Wang et al., 2024b), allowing it to generate and execute arbitrary Python scripts, run shell commands, perform file operations, and dynamically compose tool invocations. This execution flexibility allows the agent to perform compositional analyses and implement custom logic when pre-built tools are insufficient. Execution state persists across reasoning cycles through a dual-layer memory architecture. At the conversational layer,

---

**Algorithm 1** SP-Mind Skill-Augmented Agent Framework

**Require:** $Q$: User query (task specification); $\mathcal{T}$: Tool library; $\mathcal{S}$: Skill repository; $\mathcal{K}$: Keyword mappings; $\sigma$: Session state

**Ensure:** $R$: Final response with generated artifacts

1:  *// Task-Conditional Skill Injection*
2:  $s^* \leftarrow \emptyset$
3:  **for** each $(s_{\text{path}}, \text{keywords}) \in \mathcal{K}$ **do**
4:      **if** $\exists\, k \in \text{keywords}$ such that $k \subseteq Q$ **then**
5:          $s^* \leftarrow \text{LOADSKILL}(s_{\text{path}}, \mathcal{S})$
6:          **break**
7:      **end if**
8:  **end for**
9:  $P \leftarrow \text{BUILDPROMPT}(\mathcal{T}) \oplus s^*$
10: $\text{obs} \leftarrow \text{INITCONTEXT}(Q, P, \sigma)$
11: **while** not $\text{TIMEDOUT}()$ **do**
12:     *// Reason Step*
13:     $\tau \leftarrow \text{REASON}(\text{obs}, P, s^*)$
14:     **if** $\text{ISCOMPLETE}(\tau)$ **then**
15:         **return** $\text{GENERATERESPONSE}(\tau, \sigma)$
16:     **end if**
17:     *// Execution: code generation or tool call*
18:     $a \leftarrow \text{GENERATEACTION}(\tau)$
19:     $r \leftarrow \text{EXECUTEINSANDBOX}(a)$
20:     *// Observation: update state with results*
21:     $\text{obs} \leftarrow \text{obs} \cup \{(\tau, a, r)\}$
22:     $\sigma \leftarrow \text{UPDATESESSION}(\sigma, r)$
23: **end while**
24: **return** $\text{TIMEOUTRESPONSE}(\text{obs}, \sigma)$

---

session identifiers maintain dialogue history across turns, allowing the agent to reference prior reasoning and user clarifications. At the computational layer, the sandboxed execution environment preserves filesystem state, essential for multi-step computational biology tasks. Further details of our agent are provided in Appendix A.

### 3.2. Modular Spatial Proteomics Tool Integration

SP-Mind integrates established computational tools spanning the spatial proteomics analysis pipeline. Each tool is encapsulated as a Python function with standardized interfaces, enabling seamless orchestration while maintaining interoperability with underlying container-based executables (Docker, Singularity).

**Image Preprocessing.** Correcting illumination artifacts and autofluorescence prior to downstream analysis. **Tools:** BaSiC for flat-field and dark-field illumination profile estimation, enabling correction of vignetting and uneven illumination across large tissue sections (Peng et al., 2017). Backsub for pixel-wise background subtraction, scaled by exposure time, to reduce marker bleed-through and autofluo-

rescence in multi-cycle OME-TIFF images (Schapiro et al., 2022).

**Stitching & Registration.** Assembling tiled multiplexed images into seamless whole-slide mosaics and aligning multi-cycle immunofluorescence data. **Tool:** ASHLAR, a vendor-agnostic framework that executes coordinated sub-pixel registration of over $10^3$ tiles per image, ensuring single-cell accuracy across massive multi-cycle datasets (e.g., CODEX, CyCIF) (Muhlich et al., 2022).

**TMA Dearraying.** Automatically detecting and extracting individual tissue cores from tissue microarray images. **Tool:** Coreograph, a deep learning semantic segmentation framework utilizing U-Net architecture and Laplacian of Gaussian (LoG) estimators to robustly dearray highly heterogeneous samples, generating precise tissue masks via active contours (Schapiro et al., 2022).

**Cell Segmentation.** Generating single-cell and nuclear masks from multiplexed images. **Tools:** UnMicst, a 3-class U-Net trained on ∼10,400 manually annotated nuclei across 7 distinct tissue morphologies (Yapp et al., 2022), used for generating per class probability maps. S3segmenter, a versatile marker-controlled watershed framework that executes instance segmentation with puncta detection and three cytoplasm segmentation modes (Schapiro et al., 2022).

**Cell Quantification.** Extracting per-cell marker intensities and morphological features. **Tool:** MCQuant, a high-throughput extraction engine powered by scikit-image that leverages 32-bit instance masks to compute mean protein expression and comprehensive spatial topologies (nuclei and cytoplasm), generating compatible CSV feature tables (Schapiro et al., 2022).

**Clustering & Phenotyping.** Unsupervised discovery of cell populations and supervised phenotype assignment. **Tools:** Leiden (Traag et al., 2019) and Phenograph (Levine et al., 2015) algorithms for unsupervised clustering on marker expression profiles. Supervised phenotyping via rule-based classification using expert-defined marker thresholds and decision trees (Schapiro et al., 2022).

All tools support both single-image and batch processing modes, with automatic parallelization across available computational resources.

### 3.3. Spatial BioSkill Templates

It's important to note that declarative tool APIs alone are insufficient for complex scientific workflows. Reasoning about spatial proteomics queries, as well as effectively calling and chaining aforementioned tools, require advanced domain-specific knowledge. To bridge this gap, SP-Mind introduces Spatial BioSkill Templates: structured knowledge primitives encoding executable scientific methodologies as task-conditional chain-of-thought scaffolds. These skills are curated by domain experts with extensive experience in spatial proteomics workflows, encoding their procedural knowledge into reusable templates. The curation process went through 4 rounds of iterative refinement, repeatedly validated on real analytical tasks (150+). Each skill specifies reasoning strategy, prerequisite dependencies, recommended parameter heuristics, and error recovery protocols for a target spatial biology workflow, enabling the agent to execute complex pipelines with expert-level procedural guidance. To avoid context dilution, skills are dynamically injected into agent prompt based on task semantics.

## 4. SP-Bench

To bridge the critical validation gap mentioned in Section 2, we introduce SP-Bench, the first comprehensive hierarchical evaluation framework designed specifically for agentic orchestration of the end-to-end spatial proteomics analysis pipeline. SP-Bench distinguishes itself through three key pillars:

**Comprehensive Coverage:** SP-Bench spans 102 distinct natural language queries covering all 8 stages of the fluorescence-based spatial proteomics analysis pipeline and includes clustering inputs from both fluorescence-based and mass-spectrometry imaging datasets. The queries are organized into 18 task categories that reflect real-world analytical objectives such as signal calibration, artifact removal, and population profiling.

**Hierarchical Complexity:** Queries are stratified into four difficulty tiers based on pipeline depth: 40 basic queries (single stage), 28 intermediate queries (involving 2 stages), 21 advanced queries (3 stages), and 13 challenging queries (4+ stages). Each tier includes both general queries and context-specific variants requiring specific configurations such as selecting appropriate nuclear channels.

**Input Data Diversity:** The benchmark encompasses diverse tissue types, sample formats, and acquisition parameters. This diversity ensures robust evaluation across the technological heterogeneity encountered in real spatial proteomics research.

### 4.1. Benchmark Creation & Dataset

SP-Bench comprises natural language queries designed to evaluate an agent's ability to orchestrate spatial proteomics analysis pipelines. We first established 8 core pipeline stages that are essential for multiplexed tissue image analysis. **1.**

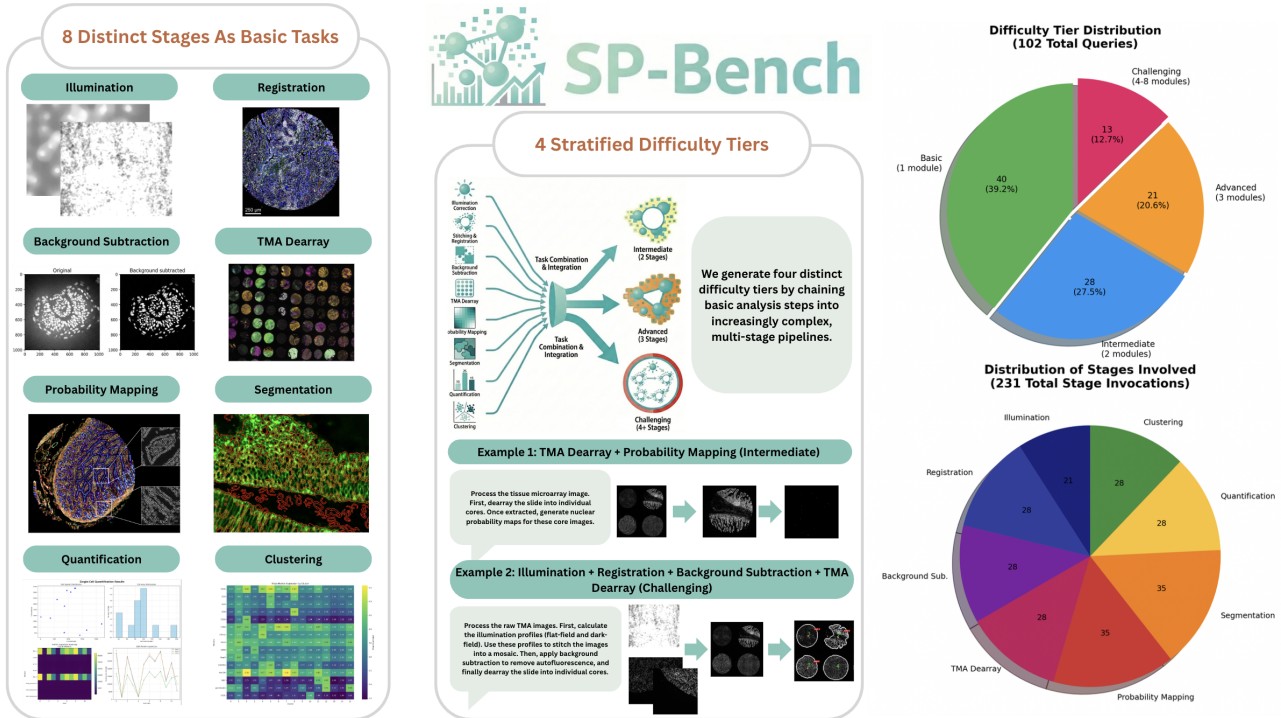

*Figure 2.* **Overview of the SP-Bench Framework and Composition.** The benchmark is constructed from eight fundamental spatial proteomics analysis stages (Left). These atomic tasks are combined to generate 3 additional stratified difficulty tiers (Center). The quantitative breakdown (Right) highlights the distribution of the 102 benchmark queries across difficulty levels and the balanced frequency of the 231 total stage invocations. Illustrative images adapted from (Schapiro et al., 2022).

**Illumination:** Correcting microscope-induced illumination intensity variations across image tiles. **2. Registration:** Stitching and aligning multi-cycle acquisitions into a unified mosaic. **3. Background Subtraction:** Eliminating tissue autofluorescence and imaging artifacts. **4. Tissue Dearray:** Identifying and extracting individual tissue cores from microarrays. **5. Probability Mapping:** Generating deep learning-based nuclear probability maps for downstream segmentation. **6. Segmentation:** Converting probability maps into discrete cell and nuclear masks via segmentation. **7. Quantification:** Quantifying per-cell marker intensities and morphological properties. **8. Clustering:** Grouping cells into biologically meaningful populations via unsupervised learning.

To systematically evaluate agent capabilities across varying levels of complexity, we organize the eight pipeline stages into four difficulty tiers comprising 18 task categories that reflect real-world spatial proteomics analytical objectives. The Basic tier (8 categories, 40 queries) evaluates mastery of stages modules in isolation. The Intermediate tier (4 categories, 28 queries) introduces two-stage workflows such as Pre-processing (illumination correction followed by stitching). The Advanced tier (3 categories, 21 queries) requires orchestrating three-stage pipelines. Finally, the Challenging tier (3 categories, 13 queries) presents complex end-to-end

workflows which requires the agent to autonomously orchestrate 4 or more stages from raw image acquisition through final cell population analysis. This hierarchical structure enables fine-grained diagnosis of agent failure modes.

For each task category, we first prompted Gemini 3 Pro to generate natural language queries that would realistically reflect how researchers might request specific analytical operations, including both general operation and context-specific configurations. All generated queries then underwent rigorous validation by human domain experts with expertise in spatial proteomics workflows. Specifically, three bioinformatics experts verified each query for: (1) biological plausibility of the requested operation and configuration, (2) correct specification of module dependencies and data flow, and (3) alignment with realistic user requests in spatial proteomics practice. Queries judged problematic by at least two of the three experts (32 out of the initial 134) were excluded from the final benchmark. Furthermore, the experts actively revised and refined queries that contained overly ambiguous instructions or incorrect workflow sequencing to ensure both quality and realism.

**Benchmark Evaluation Metric** The benchmark employs an execution accuracy metric: the percentage of queries for which the agent successfully completes the requested

analysis and fulfills the query objective. A query is counted as successful only when the agent uses the appropriate tool or code, follows all user-specified parameters, preserves the correct execution order and intermediate data flow, produces the requested outputs in the specified format and location, and completes execution without runtime errors or timeout. We further require that the agent avoid unintended modifications to input data or directory structure, provide an accurate summary of the completed operations, and transparently report any errors or warnings encountered during execution. For the full success criteria, see Appendix C.3.

**Input Data Preparation**    To ensure realistic and validated inputs for each task category, we derived our benchmark data from authoritative sources: the MCMICRO exemplar dataset (Schapiro et al., 2022), MAPS cHL dataset (Shaban et al., 2024), and PDAC IMC Dataset (Sussman et al., 2024). Our input dataset covers 5 important tissue environments (Meningioma, GI Stromal Tumor, Normal Colon, Classic Hodgkin Lymphoma, and Pancreatic Ductal Adenocarcinoma) and 4 mainstream imaging technologies (CyCIF, MIBI, CODEX, IMC).

## 5. Experiments

### 5.1. Experiment Setup

We evaluate SP-Mind using Claude Sonnet 4 as the uniform backbone LLM across all experimental conditions to ensure fair comparison. We benchmark SP-Mind (along with its no-skill version) against 5 baselines, ranging from general-purpose frameworks to specialized biomedical agents. All baselines utilize their official implementations with default configurations: **AutoGen** (Wu et al., 2024), a widely-adopted general-purpose framework for building LLM agents. We deploy it here as a single-agent baseline, excluding its multi-agent conversational features to ensure a direct comparison of reasoning abilities. **Biomni** (Huang et al., 2025), a biomedical agent framework implementing a ReAct-style reasoning loop coupled with CodeAct execution. It is natively integrated with a broad suite of standard bioinformatics tools. **ToolUniverse** (Gao et al., 2025), a comprehensive tool-augmented framework designed for scientific computing, providing extensive API coverage across general biology and data analysis domains. **Biomni-SP** and **ToolUniverse-SP**, both are domain-adapted variant of Biomni and ToolUniverse that we engineered specifically for this study. They retain the original respective agent architecture but is equipped with the spatial proteomics tool suite used by SP-Mind. We evaluate the agents on 3 complementary evaluation suites.

**SP-Bench:**    Our proposed benchmark, described in Section 4. This benchmark comprehensively evaluates agents'

abilities in spatial proteomics analysis through 102 diverse queries across 18 categories, 8 stages, and 4 difficulty tiers. Model performance is directly measured by the number of successful executions across all queries.

**CRC-CODEX Quantification Challenge:**    We benchmark the agents on their ability to execute quantification tasks on 10 high-resolution TIFF images from the CRC-CODEX (Schürch et al., 2020) dataset. The quantified results are compared to the ground truth from the original dataset using a comprehensive suite of 5 metrics. We evaluate total marker abundance via Pearson and Spearman correlations, phenotype distribution fidelity using Maximum Mean Discrepancy (MMD) (Gretton et al., 2012), and biological co-expression preservation via correlation matrix similarity. We also employ a grid-based spatial Spearman correlation to verify signal localization without requiring strict cell-to-cell mapping. These measures are designed to be segmentation-tolerant, ensuring that agents are not overly penalized for making segmentation choices that differ from the ground truth while still capturing the correct biological signal through quantification.

**Cell Annotation Challenge:**    In this suite, we benchmark the agents on annotating cells based on marker expression data. We utilize 4 large-scale CSV datasets from MAPS cHL (Shaban et al., 2024) and PDAC IMC (Sussman et al., 2024), covering ≈2 million cells, to measure the agents' annotation accuracy. To evaluate performance, we utilize CyteOnto (Ahuja et al., 2025) instead of naive string matching. CyteOnto is a semantic similarity framework that maps labelled and ground-truth labels to Cell Ontology terms using text embeddings of LLM-augmented descriptions. It derives a kernel-induced similarity score (ranging from 0 to 1) to quantify accuracy, ensuring that semantic relationships and biological meaning are preserved.

### 5.2. Quantitative Analysis

**SP-Bench**    As shown in Table 1, SP-Mind achieves consistent state-of-the-art performance (68.9% average) across all four difficulty tiers, marking a significant improvement over comparable agent baselines. We observe a clear performance hierarchy where domain-specialized agents with spatial proteomics toolkit dramatically outperform generalist agents. While generalist models such as Biomni (19.3%), ToolUniverse (15.0%), and AutoGen (14.7%) struggle to execute complex spatial queries—collapsing to near-zero performance in the Advanced and Challenging tiers—SP-Mind maintains robustness throughout. Furthermore, SP-Mind outperforms the strongest specialized baseline, Biomni-SP (55.4%), by a substantial margin. This gap is particularly significant in the Advanced tier, where SP-Mind (61.9%) nearly doubles the success rate of Biomni-SP (36.5%), demon-

*Table 1.* **Agent Performance on SP-Bench.** The benchmark comprises 102 distinct spatial proteomics queries spanning 18 categories and 8 analysis stages. Evaluation is stratified by difficulty: Basic (40 queries, 8 categories), Intermediate (28, 4), Advanced (21, 3), and Challenging (13, 3). Scores report execution success rates (%). SP-Mind significantly outperforms both generalist agents and domain-specific variants across all difficulty tiers. We run the evaluation 3 times, with mean and standard deviation reported.

| Difficulty | AutoGen | Biomni | ToolUniverse | Biomni-SP | ToolUniverse-SP | SP-Mind (no skill) | SP-Mind |
|---|---|---|---|---|---|---|---|
| Basic | $31.7 \pm 3.8$ | $37.5 \pm 5.0$ | $30.8 \pm 3.8$ | $88.3 \pm 1.4$ | $90.8 \pm 1.4$ | $95.0 \pm 2.5$ | $\mathbf{95.8 \pm 1.4}$ |
| Intermediate | $23.8 \pm 2.1$ | $28.6 \pm 0.0$ | $21.4 \pm 3.6$ | $73.8 \pm 4.1$ | $70.2 \pm 7.4$ | $82.1 \pm 7.1$ | $\mathbf{84.5 \pm 7.4}$ |
| Advanced | $3.2 \pm 5.5$ | $11.1 \pm 2.7$ | $7.9 \pm 2.7$ | $36.5 \pm 5.5$ | $25.4 \pm 7.3$ | $44.4 \pm 7.3$ | $\mathbf{61.9 \pm 4.8}$ |
| Challenging | $0.0 \pm 0.0$ | $0.0 \pm 0.0$ | $0.0 \pm 0.0$ | $23.1 \pm 0.0$ | $15.4 \pm 7.7$ | $28.2 \pm 4.4$ | $\mathbf{33.3 \pm 4.4}$ |
| Average | 14.7 | 19.3 | 15.0 | 55.4 | 50.5 | 62.4 | **68.9** |

*Table 2.* **Agent Performance on CRC-CODEX Quantification Challenge.** Evaluation is conducted using segmentation-tolerant metrics to compare agent quantification against Ground Truth. Each agent quantified the same dataset (10 images) 3 times; values reported are the mean and standard deviation across trials. Key metrics include Pearson $r$ and Spearman $\rho$ for marker abundance correlation, Correlation Matrix similarity for biological relationship preservation, and Maximum Mean Discrepancy (MMD) for phenotype population distance (lower is better). Generalist baselines (Biomni, ToolUniverse, AutoGen) are excluded as their produced outputs significantly fall short of the expectation of the task (e.g., largely incomplete csv, running grading script is not possible).

| Agent | Pearson $r$ | Spearman $\rho$ | Corr Matrix | MMD | Spatial Spearman |
|---|---|---|---|---|---|
| Biomni-SP | $0.940 \pm 0.044$ | $0.870 \pm 0.146$ | $0.703 \pm 0.097$ | $0.390 \pm 0.110$ | $0.460 \pm 0.112$ |
| ToolUniverse-SP | $0.929 \pm 0.068$ | $0.865 \pm 0.156$ | $0.711 \pm 0.096$ | $0.393 \pm 0.123$ | $0.482 \pm 0.107$ |
| SP-Mind (no skill) | $0.945 \pm 0.038$ | $0.892 \pm 0.132$ | $0.725 \pm 0.096$ | $0.370 \pm 0.097$ | $0.497 \pm 0.112$ |
| SP-Mind | $\mathbf{0.953 \pm 0.028}$ | $\mathbf{0.902 \pm 0.123}$ | $\mathbf{0.727 \pm 0.110}$ | $\mathbf{0.368 \pm 0.094}$ | $\mathbf{0.513 \pm 0.105}$ |

*Table 3.* **Agent Performance on the Cell Annotation Challenge.** Evaluation is conducted across 4 datasets encompassing $\approx$2 million cells. Each agent annotates each dataset 3 times. We report the CyteOnto GHK similarity score (Ahuja et al., 2025) averaged across all cells and all trials, which quantifies the semantic similarity between agent and ground-truth annotations, along with its standard deviation. This metric maps labels to the Cell Ontology space using Qwen3-Embedding-8B model (Yang et al., 2025) and applies a Gaussian Heat Kernel with $\sigma = 0.25$ to penalize semantic drift. ASTIR (Geuenich et al., 2021) is included as a specialized ML-based baseline. Generalist baselines (Biomni, ToolUniverse) are excluded as their cell-annotation capabilities are subsumed by their SP-specific variants.

| Dataset | AutoGen | Biomni-SP | ToolUniverse-SP | ASTIR | SP-Mind (no skill) | SP-Mind |
|---|---|---|---|---|---|---|
| cHL_1_MIBI | $0.567 \pm 0.113$ | $0.606 \pm 0.088$ | $0.602 \pm 0.049$ | $0.339 \pm 0.000$ | $0.534 \pm 0.054$ | $\mathbf{0.630 \pm 0.021}$ |
| cHL_2_MIBI_5 | $0.616 \pm 0.052$ | $0.663 \pm 0.045$ | $0.655 \pm 0.075$ | $0.258 \pm 0.000$ | $0.672 \pm 0.064$ | $\mathbf{0.765 \pm 0.058}$ |
| cHL_CODEX | $0.668 \pm 0.068$ | $0.599 \pm 0.090$ | $0.668 \pm 0.020$ | $\mathbf{0.738 \pm 0.000}$ | $0.663 \pm 0.025$ | $0.682 \pm 0.045$ |
| PDAC_IMC | $0.592 \pm 0.139$ | $0.706 \pm 0.019$ | $0.693 \pm 0.042$ | $0.426 \pm 0.000$ | $\mathbf{0.762 \pm 0.063}$ | $0.648 \pm 0.040$ |
| Average | 0.610 | 0.643 | 0.654 | 0.440 | 0.657 | **0.681** |

strating superior capability in handling multi-step complex analysis.

**CRC-CODEX Quantification Challenge** Table 2 details agent performance, where SP-Mind obtains the best mean score across all five segmentation-tolerant metrics. SP-Mind achieves the highest alignment with Ground Truth in global marker abundance, securing a Pearson $r$ of 0.953 and a Spearman $\rho$ of 0.902, indicating exceptional linear and rank-order precision. Beyond simple abundance, SP-Mind demonstrates a superior ability to preserve high-dimensional biological heterogeneity, attaining the lowest Maximum Mean Discrepancy (MMD) score of 0.368 and the highest Correlation Matrix similarity (0.727). Notably, in the grid-based Spatial Spearman metric—which penalizes correct aggregate counts if they appear in the wrong image location—SP-Mind (0.513) outperforms the nearest competitor, ToolUniverse-SP (0.482), by a meaningful margin. This performance suggests that SP-Mind's autonomous selection of tool parameters and segmentation strategies aligns more closely with expert scientific practices, allowing it to capture complex marker co-expression patterns and phenotype population structures that baselines miss.

**Cell Annotation Challenge** Table 3 summarizes agent performance on the Cell Annotation Challenge. SP-Mind achieves the highest average CyteOnto GHK similarity score (0.681) across the four datasets, outperforming all agentic

baselines and the specialized ASTIR model. The gains are most pronounced on the two MIBI datasets: SP-Mind reaches 0.630 on cHL_1_MIBI and 0.765 on cHL_2_MIBI_5, exceeding the strongest competing agent by 0.024 and 0.102, respectively. The results also reveal complementary strengths and limitations across methods. ASTIR, the specialized ML baseline, excels on cHL_CODEX (0.738), but it fails to generalize to other modalities, dropping to significantly lower scores elsewhere, likely due to sensitivity to feature sparsity. In contrast, SP-Mind provides the best overall balance across datasets, ranking first on both MIBI tasks and remaining competitive on cHL_CODEX. However, on PDAC_IMC, SP-Mind (0.648) underperforms Biomni-SP (0.706), ToolUniverse-SP (0.693), and the no-skill ablation (0.762), indicating that the current skill templates may not transfer uniformly across all imaging modalities.

### 5.3. Qualitative Analysis

Extensive analysis of execution logs reveals that the performance gap between SP-Mind and generalist agents stems primarily from a lack of methodological foresight rather than coding incapacity. Generalist agents frequently struggle with the tacit constraints of spatial biology such as physical stage drift in stitching, often reverting to naive algorithmic reinvention. SP-Mind's skill-augmented architecture effectively bridges this gap, acting as a cognitive guardrail that enforces expert protocols and ensures scientific fidelity. The evaluation further reveals that SP-Mind effectively bridges the gap between the brittleness of specialized ML models and the imprecision of generalist agents in cell annotation. While specialized baselines like ASTIR falter on sparse modalities (MIBI), SP-Mind leverages semantic plasticity and domain-specific methods to resolve subtle phenotypic states. We illustrate these qualitative findings through two representative case studies (Figure 3).

**Image Stitching & Registration Case Study** This task requires stitching the MCMICRO exemplar-002 dataset (10 cyclic OME-TIFF acquisitions, each containing a $5 \times 6$ mosaic of tiles) into a seamless tissue map stored as an OME-TIFF. The baseline agent, Biomni, attempted to write a custom stitching algorithm from scratch. While it successfully parsed the metadata to determine the $5 \times 6$ grid layout, it failed to perform actual image registration. Instead, it implemented a "blind stitching" approach, placing tiles based on theoretical stage coordinates with a hard-coded 10% overlap. This ignores mechanical stage drift, resulting in alignment artifacts. Furthermore, it struggled with OME-XML encoding and created separate files for pyramid levels rather than a single multi-resolution file. In contrast, SP-Mind correctly called the domain-specific ASHLAR tool. When initially faced with a parameter error, the agent engaged in active reflection, inspecting the function signature

to correct the input arguments. It successfully executed phase-correlation based registration to quantify and correct sub-pixel stage drift (calculating offsets of $\approx 41$ pixels), producing a scientifically accurate, fully aligned OME-TIFF in fewer interaction turns.

**Cell Annotation (cHL_2_MIBI_5) Case Study** This task involves diagnosing cell identities across 12 clusters in a classical Hodgkin lymphoma dataset using 46 protein markers. The baseline agent, Autogen, relies on raw absolute expression means; consequently, ubiquitous structural channels (e.g., Histone H3) dominate its top-marker evidence. This obscures discriminative lineage markers, leading to lineage collapse (e.g., annotating the ground-truth dendritic-cell and NK clusters as generic "T cells") and loss of subtype structure (merging M1 and M2 into a single "Macrophages" label). In contrast, SP-Mind employs a domain-specific rank/enrichment analysis to normalize signal-scale variation and down-weight non-informative markers, thereby recovering distinct lineages (DC, NK, CD8 T). Furthermore, SP-Mind distinguishes subtle phenotypic states such as macrophage polarization, separating an M2-like CD163$^{\text{hi}}$ compartment from a more inflammatory M1-like state (characterized by CD68$^{+}$ and Granzyme B expression), demonstrating the model's ability to emulate expert biological reasoning through statistical ranking and hierarchical logic.

**Failure Case Analysis** Despite SP-Mind's overall robustness, our execution traces reveal several recurring failure modes in complex multi-stage workflows. A representative failure mode arises from file-management and state-tracking errors. When the workflow is long, after successfully completing several upstream stages, the agent may save an intermediate artifact to an incorrect directory or pass a stale file path to a downstream tool. A second failure mode occurs during background subtraction. Unlike several other pipeline stages that expose relatively standardized inputs, background subtraction can require additional input engineering before execution, such as adapting a marker metadata file into the format expected by the underlying tool. When background subtraction is requested as an isolated task, SP-Mind usually inspects the required schema, modifies the metadata file, and completes the operation successfully. However, in longer chained workflows, the agent may under-investigate formatting errors. These observations suggest that autonomous spatial proteomics agents require not only stronger biological reasoning, but also more rigorous bookkeeping mechanisms for managing files, intermediate artifacts, and tool-specific input contracts across extended analytical pipelines.

**User Query**

Register the multi-tile microscopy images found in {{INPUT_DIR}} to create a seamless tissue map. The final output should be saved as a pyramidal OME-TIFF at {{OUTPUT_DIR}}

**Biomni AutoGen**

**Biomni Baseline**
- **Metadata-First Parsing:** Walked the directory, read OME-XML, and recovered a full 5×6 tile grid from per-plane (X,Y) stage positions.
- **Blind Grid Placement:** Implemented custom stitching by placing tiles onto a canvas with a hard-coded 10% overlap, no correlation-based alignment or drift correction.
- **Pyramid Workaround:** Failed to emit a true single-file pyramidal OME-TIFF and instead exported downsampled pyramid levels as separate TIFFs.

**SP-Mind**

**SP-Mind (Ours)**
- **Tool-Driven Registration:** Immediately invoked the stitching+registration routine (ASHLAR) rather than implementing stitching logic from scratch.
- **Active Reflection:** On a parameter error, inspected the function signature and corrected the script accordingly.
- **Drift-Corrected Output:** Completed end-to-end registration and stitching, producing a single registered OME-TIFF and reporting an estimated drift offset of ≈41 pixels in few turns.

Perform cell type annotation on the single-cell data cHL_2_MIBI_5.csv. The data contains a 'cluster' column with numeric cluster IDs and multiple protein marker columns. Assign biologically meaningful cell type labels for the csv. Add only a single new column called 'annotation' to the output. Each unique cluster ID must correspond to exactly one annotation label.

**Autogen Baseline**
- **Fragmented Execution:** Trajectory was interrupted by state-loss errors (NameError), forcing the agent to repeatedly reload data and restart analysis.
- **Raw Intensity Thresholding:** Relied on absolute mean expression values for inference.
- **Signal Masking:** High-intensity structural markers (e.g., Histone H3) dominated the top-k feature lists, obscuring lower-intensity lineage markers and leading to generic, collapsed annotations.

**SP-Mind (Ours)**
- **Statistical Profiling:** First executed exploratory scripts to determine data shape and marker availability.
- **Rank-Based Algorithm:** Instead of using raw values, generated a custom Python script to rank clusters by marker expression, normalizing batch effects and signal intensity differences.
- **Hierarchical Logic:** Applied combinatorial rules to the rankings (e.g., identifying a Treg-enriched cluster via CD4 and FoxP3 relative enrichment), successfully resolving granular subtypes.

*Figure 3.* Comparison of agent behaviors on two spatial biology tasks. Top: Image Stitching & Registration. Bottom: Cell Annotation.

## 6. Conclusion

We presented SP-Mind, an agent that bridges the gap between generalist reasoning and domain-specific execution in spatial proteomics. A primary limitation of the current framework is its reliance on human-curated skill templates; the agent cannot yet autonomously formalize novel workflows into reusable skills. Future work will address this by implementing autonomous skill synthesis, enabling the agent to crystallize successful execution traces into new procedural templates. This capability will close the loop between active exploration and knowledge consolidation, advancing agents to continuously learning scientific assistants.

## Impact Statement

This paper presents work whose goal is to advance the field of Machine Learning. There are many potential societal consequences of our work, none which we feel must be specifically highlighted here.

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

# A. SP-Mind Core Implementation Details

This appendix provides a detailed exposition of the SP-Mind operational framework. The system implements a skill-augmented ReAct (Reasoning and Acting) loop that enables dynamic tool orchestration for complex spatial proteomics workflows. The agent navigates workflows through iterative cycles of reasoning, code execution, and observation—guided by task-conditional procedural knowledge.

## A.1. The Skill-Augmented ReAct Cycle

SP-Mind extends the foundational ReAct paradigm (Yao et al., 2022) with structured procedural knowledge injection. The agent iterates through three core phases:

- **Observe** Ingest user task specifications and execution outputs from prior actions—including tool return values, file contents, terminal outputs, and error traces. Following a data-first protocol, the agent is instructed to inspect data structures and distributions (via summary statistics and print operations) before committing to analytical parameters.

- **Reason:** Analyze the current observations against the injected Skill Template. The template provides structured scaffolding (prerequisite checks and parameter heuristics) that attempts to prevent the LLM from hallucinating invalid workflow steps.

- **Execute:** Generate and run executable code within a managed environment. Unlike structured tool-calling approaches that limit agents to pre-defined API endpoints, SP-Mind utilizes the Claude Code CLI interface to compose arbitrary Python scripts. This allows for on-the-fly data transformation and logic patching essential for scientific computing.

## A.2. Architecture

SP-Mind relies on four interconnected components to execute the skill-augmented ReAct cycle:

### A.2.1. REASONING ENGINE

SP-Mind employs the Claude Sonnet-series (selectable) as the backbone LLM, leveraging its code execution capabilities through the official `claude-agent-sdk`. The LLM functions as the central orchestrator, maintaining multi-turn state via the SDK's session management. SP-Mind employs a composite system prompt strategy that dynamically assembles:

1. **Environment Context:** Working directory.

2. **Tool Definitions:** Signatures for the spatial proteomics pipeline tools (e.g., `unetcoreograph`, `s3segmenter`).

3. **Data-First Constraints:** Instructions to explore data distributions prior to analysis.

### A.2.2. SKILL TEMPLATE REPOSITORY

An expert curated collection of structured methodology templates encoding expert procedural knowledge. Skills are indexed by task-type keywords (mapped in `SKILL_KEYWORDS`). When a specific task (e.g., "clustering," "stitching") is detected in the user query, the relevant Markdown-based skill is retrieved and injected directly into the system prompt. This ensures the agent receives focused procedural guidance (prerequisites, parameter heuristics, and error recovery protocols) without context window dilution. Below is an example skill template for cell annotation.

**Cell Type Annotation Skill Template**

```
# Cell Type Annotation Skill

## Analysis Methodology

### 1. Use Rank-Based Analysis
- Do NOT rely solely on absolute expression values per cluster
- For each marker, calculate its **ranking** across all clusters
- Focus on which clusters rank highest for specific markers
- Even with low absolute values, the cluster ranking #1 for a marker may still
    represent that cell type

### 2. Consider Marker Combination Patterns
- Do NOT make annotations based on a single highly-expressed marker
- Consult your knowledge of classic marker combinations for each cell type
- Consider both **positive markers** AND **negative markers** (exclusion markers)

### 3. Identify Each Cluster's Unique Markers
- For each cluster, find markers that are **most unique relative to other clusters**
- Identify which markers rank #1 or #2 in that cluster
- Look for markers where a cluster is significantly higher than all others
- These unique markers are the strongest evidence for cell type identity

### 4. Be Aware of Panel Limitations
- The marker panel may lack classic markers for certain cell types
- If a key marker is missing, use alternative lineage markers
- For clusters that are difficult to clearly determine, use generic labels like "Other"
    or "Unknown"

### 5. Workflow Summary
1. Load data and calculate mean expression per cluster
2. Calculate rankings: for each marker, rank clusters from highest to lowest
3. For each cluster, identify its top-ranked markers
4. Match marker combinations to known cell type signatures
5. Verify with negative markers (what the cluster does NOT express)
6. Assign annotations based on the combined evidence

## Output Requirements

In addition to the full annotated CSV file, you MUST also save a summary file called `
    annotation_summary.csv` in the same output directory, which should contain:

- **Cluster**: The cluster ID (integer)
- **Annotation**: The assigned cell type label

This makes it easy to quickly verify the annotation results.
```

## A.2.3. Tool Execution Layer (Container Abstraction)

To ensure reproducibility across computing environments (local workstations vs. HPC clusters), SP-Mind abstracts the execution logic from the runtime environment. The spatial proteomics tool suite automatically detects and utilizes the available container runtime—Docker, Apptainer, or Singularity—allowing the agent to invoke complex binary dependencies (e.g., Ashlar, UnMicst) through uniform Python wrappers without managing low-level container flags.

SP-Mind operationalizes the container abstraction through a "Python-to-Container Bridge" pattern that encapsulates complex CLI dependencies within Python signatures. The wrapper functions perform critical orchestration logic beyond simple command execution. First, they implement automatic path resolution, dynamically calculating the common parent directory of input/output files to establish valid bind mounts regardless of the host OS file structure. Second, they enforce parameter consistency checks prior to container instantiation, preventing runtime failures caused by invalid flag combinations. Finally, rather than returning raw exit codes, the wrappers capture standard streams (stdout/stderr) to construct a semantic research

log. This text-based return object summarizes file sizes, execution process, and error traces, providing the LLM with a comprehensive observation space to reason about the tool's success or failure.

### A.2.4. STATE MANAGER

State persistence operates at two layers. At the conversational layer, the SDK's session management (`session_id`) maintains dialogue history across turns, enabling the agent to reference prior reasoning without redundant context re-injection. At the computational layer, the sandboxed execution environment preserves filesystem state—generated segmentation masks, quantification tables, and diagnostic plots persist across reasoning cycles, allowing subsequent queries to build upon prior computational artifacts. This dual-layer architecture is essential for iterative scientific workflows where multi-step analyses span extended user interactions.

### A.2.5. FULL PROMPT ARCHITECTURE

SP-Mind uses a modular prompting scheme composed of: (1) a fixed base system prompt defining the agent role, execution environment, available tools, and data-first reasoning protocol; (2) standardized tool descriptions and usage guidance; and (3) a dynamically injected Spatial BioSkill Template selected based on the user query/task type. Thus, the full prompt architecture is: [Base prompt] + [tool specifications] + [task-relevant skill template] + [user query]. The base prompt is shown below.

## Base System Prompt

```
## You are SP-Mind, an expert spatial proteomics data analysis assistant.

## Your Environment
- Working directory: {self.path}

## Available SP-Mind Tool Modules
{tool_modules_str}

## How to Use SP-Mind Tools

### Step 1: Check function signature before using
// ... example code ...

### Step 2: Import and use the function
// ... example code ...

For longer scripts, write to a file first and then execute it.

## Container Runtime Note
SP-Mind tools auto-detect and use available container runtime (Docker, Apptainer, or
    Singularity).
- On Mac/local: Docker is preferred (start Docker Desktop if needed)
- On HPC cluster: Apptainer/Singularity is preferred
All tools accept `container_runtime` parameter: "auto" (default), "docker", "apptainer
    ", "singularity"

## CRITICAL: Data-First Approach

When analyzing data (especially for tasks like cell type annotation, clustering, or
    quantification):

1. **EXPLORE DATA FIRST** - Before writing any analysis code:
   - Load the data and print its structure
   - Calculate summary statistics (mean, std, range) for each group/cluster
   - Print the actual values to understand the data distribution
   - Examine what markers/features are high or low for each group

2. **MAKE DATA-DRIVEN DECISIONS** - Base your analysis parameters on observed values:
   - Don't use generic thresholds (like 0.3) without checking the data
   - Look at actual expression ranges before setting cutoffs
   - Examine marker combinations, not just single markers

3. **ITERATE AND REFINE** - Don't write one big script:
   - Execute small code blocks and print results
   - Adjust your approach based on what you observe
   - Re-examine ambiguous cases

// ... example workflow ...

## Instructions
1. Analyze the user's biomedical task
2. **Explore and understand the data first** by printing intermediate results
3. Write and execute Python code using SP-Mind tools
4. Print results clearly and save plots to files
5. Explain your findings

Solve the user's task step by step using the available tools.
```

# B. Algorithm Execution Flow

This section provides a step-by-step exposition of SP-Mind's execution pipeline.

### B.1. Step 1: Initialization and Environment Preparation

The agent initialization establishes the execution environment through sequential operations:

1. **Configuration Loading:** The agent accepts parameters for the working directory, timeout, and permission limits.

2. **Tool Registry Loading:** The `read_module2api()` function parses function signatures and docstrings from the tool directory into a structured dictionary, enabling the agent to programmatically discover available tools.

3. **Session State Initialization:** The session identifier (`session_id`) is initialized to `None`, marking the start of a fresh conversation.

### B.2. Step 2: Task-Conditional Skill Retrieval

Upon invoking `go(prompt)`, the agent executes `_retrieve_skill(task)`:

- **Keyword Matching:** The prompt is normalized and compared against the `SKILL_KEYWORDS` dictionary. For example, the keyword "quantif" triggers the loading of `quantification/cell_quantification.md`.

- **Graceful Fallback:** If no keywords match, the agent proceeds with the base system prompt (general tool documentation only), avoiding the injection of irrelevant scaffolding.

### B.3. Step 3: System Prompt Assembly

The `build_system_prompt_with_skill` function constructs the composite system prompt by concatenating the Base Prompt (environment context, tool signatures, and data-first protocols) with the Active Skill. If a skill is retrieved, it is appended under the header:

```
## ACTIVE SKILLS (Follow This Methodology!)
[Skill content]
```

This explicit demarcation instructs the LLM toward the retrieved procedure while retaining full access to the underlying tool library.

### B.4. Step 4: SDK Options Construction

The `build_options` function configures the `ClaudeAgentOptions` object differently for new versus resumed sessions:

- **New Session:** Includes the fully assembled `system_prompt` containing the injected skill.

- **Resumed Session:** Omits the `system_prompt` parameter and instead passes `resume=self._session_id`. This adheres to the SDK protocol where the conversation context (including the initial skill and tool definitions) is persisted server-side.

### B.5. Step 5: Asynchronous Query Execution

The `run_async` coroutine orchestrates the execution loop via the Claude Agent SDK. Unlike the high-level description in Appendix A, this step handles specific SDK message types:

- `SDKSystemMessage`: Captures the new `session_id` for persistence.

- `AssistantMessage`: Processes the LLM's output blocks, including `ThinkingBlock` (chain-of-thought) and `ToolUseBlock` (structured command generation).

- `SDKResultMessage`: Signals loop termination and provides usage statistics (cost and turn count).

### B.6. Step 6: Session Persistence and State

Upon loop completion, the agent preserves the `session_id`. Generated artifacts (segmentation masks, plots) persist in the filesystem, remaining accessible for subsequent analysis.

### B.7. Step 7: Multi-Turn Interaction Pattern

For follow-up queries, the agent detects the active session and sets `is_resume=True`. It resumes the SDK session, inheriting the prior conversation history. This enables iterative refinement (e.g., adjusting parameters based on previous plots) without re-initializing the environment.

### B.8. Error Recovery and Robustness

SP-Mind implements multi-layered error handling to maintain workflow stability:

- **Execution-Level Signaling:** The SDK's message stream includes explicit `is_error` flags in `SDKResultMessage` objects. These signals allow the agent to distinguish between successful execution with zero results and genuine runtime failures.

- **Observation-Driven Recovery:** Rather than relying on hardcoded retry logic, SP-Mind leverages the ReAct loop's intrinsic adaptability. When a tool fails, the wrapper captures the full `stderr` trace and returns it as a textual observation. The LLM analyzes this diagnostic feedback to identify root causes (e.g., parameter constraints, missing dependencies) and attempts remediation, such as adjusting thresholds or selecting alternative algorithms, in the subsequent reasoning turn.

- **Transparent Runtime Fallback:** To ensure execution across heterogeneous environments, tool wrappers implement automatic container detection. If the preferred runtime (e.g., Docker) is unavailable, the system transparently degrades to Apptainer or Singularity without user intervention, preventing infrastructure-level failures from halting the analytical pipeline.

### B.9. Output Formatting and Artifact Management

SP-Mind structures outputs to support both programmatic consumption and human interpretability:

- **The "Research Log" Pattern:** Tool wrappers are designed to return structured text summaries rather than raw exit codes. These logs aggregate execution metadata (e.g. timestamps), file manifests (paths, sizes), and validation checks (e.g., "15,432 cells identified"). This verbose observation space provides the LLM with the context necessary to verify success and hallucination-check its own outputs.

- **Artifact Persistence:** Generated files (segmentation masks, quantification tables, diagnostic plots) are saved to the sandboxed working directory (`self.path`). Because the session context persists, these artifacts remain addressable by file path across multi-turn interactions, allowing the user to refine visualizations or request downstream analysis on previously generated data.

- **Structured Final Response:** The system prompt directs the agent to synthesize a terminal response containing a natural language summary of biological findings, accompanied by an explicit manifest of generated files. This ensures the user receives both the analytical insight and the location of the raw data products.

## C. SP-Bench: Comprehensive Benchmark Details

### C.1. Complete Category Taxonomy Table

Table 4 details the complete taxonomy of the benchmark, categorized by difficulty tier. It outlines the specific stages involved in each category and the number of queries included in the benchmark dataset.

*Table 4.* Complete Category Taxonomy Table

| Tier | Category | Stages Involved | Queries |
|------|----------|-----------------|---------|
| **Basic** | Signal Calibration | Illumination | 5 |
| | Image Registration | Registration | 5 |
| | Artifact Removal | Background Subtraction | 5 |
| | Tissue Segmentation | TMA Dearray | 5 |
| | Probability Mapping | Probability Mapping | 5 |
| | Cell Segmentation | Segmentation | 5 |
| | Feature Extraction | Quantification | 5 |
| | Phenotypic Analysis | Clustering | 5 |
| **Intermediate** | Pre-processing | Illumination → Registration | 7 |
| | Core-level Segmentation | TMA Dearray → Probability Mapping | 7 |
| | Cell Segmentation Pipeline | Probability Mapping → Segmentation | 7 |
| | Population Profiling | Quantification → Clustering | 7 |
| **Advanced** | TMA Post-processing | Registration → Background Sub. → TMA Dearray | 7 |
| | High-fidelity Segmentation | Background Sub. → Prob. Mapping → Segmentation | 7 |
| | Downstream Interpretation | Segmentation → Quantification → Clustering | 7 |
| **Challenging** | TMA Full Pipeline | Illumination → Registration → Background Sub. → TMA Dearray | 4 |
| | Challenging Pipeline | Prob. Mapping → Segmentation → Quantification → Clustering | 4 |
| | Grand Pipeline | All 8 stages | 5 |

## C.2. Example Queries for Basic Tasks

Below we provide one representative example query for each of the eight basic pipeline stages. Each example demonstrates realistic natural language phrasing that a researcher might use when interacting with an agentic system.

1. **Illumination**

   "Generate the illumination correction profiles for the 10 cycles of raw OME-TIFF images located in `/data/raw_images/`. All resulting flat-field and dark-field profiles should be stored in `/data/illumination_profiles/` for use in the subsequent stitching step."

2. **Registration**

   "Stitch the 10 separate raw OME-TIFF cycle files located in `/data/raw_cycles/` into a single multi-channel mosaic. Ensure the cycles are registered correctly and save the output to `/data/registered/`."

3. **Background Subtraction**

   "Perform background subtraction on the stitched OME-TIFF image located at `/data/registered/mosaic.ome.tif` to remove tissue autofluorescence. Use the channel metadata provided in `/data/markers.csv` and save the corrected image to `/data/background_corrected/`."

4. **TMA Dearray**

   "Take the stitched TMA image located at `/data/tma_scan.ome.tif` and dearray it into individual tissue cores. Store the resulting cropped core images and the QC map in `/data/cores/`."

5. **Probability Mapping**

"Generate nuclear probability maps for the stitched whole-slide image located at `/data/tissue_scan.ome.tif`. Save the resulting nuclei foreground and contour maps into `/data/probability_maps/` to facilitate downstream segmentation."

6. **Segmentation**

"Perform watershed-based segmentation on the whole-slide image located at `/data/tissue_scan.ome.tif`. Use the nuclear probability maps stored in `/data/probability_maps/` to generate the final cell masks. Save all results, including nuclei and quality control outlines, to `/data/segmentation/`."

7. **Quantification (Feature Extraction)**

"Perform single-cell quantification on the stitched image located at `/data/tissue_scan.ome.tif`. Use the cell segmentation mask found at `/data/segmentation/cell_mask.tif` and the marker names listed in `/data/markers.csv`. Save the resulting quantification CSV to `/data/quantification/`."

8. **Clustering (Phenotypic Analysis)**

"Perform unsupervised clustering on the single-cell quantification data located at `/data/quantification/cells.csv`. Use the Leiden algorithm to identify distinct cell populations and also save UMAP plots and clustered data to `/data/clustering/`."

## C.3. Evaluation Criteria for Query Success

A query is strictly marked as **successful** only if it satisfies all of the following nine verification criteria. This rigorous evaluation ensures that the agent not only produces the correct scientific output but also adheres to operational constraints and best practices in computational reproducibility.

1. **Tool and Code Correctness:** The agent must invoke the scientifically appropriate tool(s) or generate syntactically and logically correct code that directly addresses the specific demands of the task.

2. **Parameter Adherence:** All parameters explicitly specified in the user prompt (e.g., channel names, thresholds, algorithms) must be correctly applied. Deviations from specified parameters result in failure.

3. **Sequential Ordering and Data Flow:** For multi-stage queries, the agent must execute stages in the correct logical order. Furthermore, the output of one stage must be correctly passed as the input to the subsequent stage (correct intermediate data flow).

4. **Output Fidelity:** The generated output files (images, CSVs, plots) must functionally satisfy the query's demand. This includes matching file formats, dimensions, data types, and containing all specific features requested by the query. In our evaluation, this was manually assessed by three bioinformatics experts, who verified that the generated outputs were biologically valid and contained sufficient information suitable to be passed to the next workflow stage.

5. **Output Location Compliance:** All final results must be saved exactly to the output directory specified in the prompt. Saving files to default or temporary locations without moving them to the requested path is considered a failure, as biological batch processing has complex directory structures.

6. **Directory Integrity:** The agent must not modify, delete, or overwrite the original input data. Additionally, the structure of the parent output directory should remain consistent, except for the creation of the requested results.

7. **Execution Completion:** The tool or generated code must run to completion without runtime errors, crashes, or exceeding the designated (30 min) timeout period.

8. **Summary Accuracy:** Upon completion, the agent must provide a natural language summary that accurately reflects the operations performed. Hallucinating steps that were not executed results in failure.

9. **Error Transparency:** If an error or warning is encountered during execution (even if recoverable), the agent must correctly report it to the user rather than suppressing it or claiming a false success.

## C.4. Input Dataset Details

The SP-Bench input data was curated to provide validated, realistic inputs for each pipeline stage while ensuring consistent data provenance.

PRIMARY DATASET (STAGES 1–7)

The majority of SP-Bench queries (Illumination through Quantification) utilize data derived from the **MCMICRO Exemplar-002 Dataset** (Schapiro et al., 2022). The exemplar dataset natively provides raw inputs for the illumination and registration stages.

To generate validated inputs for downstream stages (background subtraction, TMA dearray, probability mapping, segmentation, and quantification), domain experts manually executed the complete MCMICRO workflow, producing ground-truth intermediate outputs at each step.

CLUSTERING DATASET (STAGE 8)

For the clustering stage, we selected four cell quantification CSVs from independent spatial proteomics studies to ensure diversity in data distribution:

- cHL_1_MIBI and cHL_2_MIBI_5: Classic Hodgkin Lymphoma samples acquired via MIBI (Shaban et al., 2024).

- cHL_CODEX: Classic Hodgkin Lymphoma sample acquired via CODEX (Shaban et al., 2024).

- PDAC_IMC: Pancreatic Ductal Adenocarcinoma sample acquired via IMC (Sussman et al., 2024).

To ensure computational tractability while preserving biological heterogeneity, CSVs exceeding 100 MB were subsampled using stratified random sampling that maintains equal representation across all cell clusters.

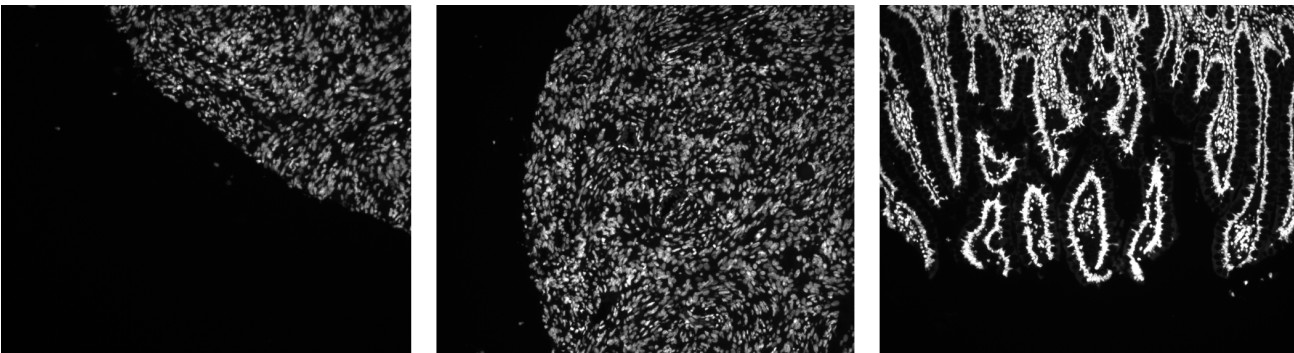

*Figure 4.* Example Input of Illumination & Registration (raw unstitched OME-TIFF)

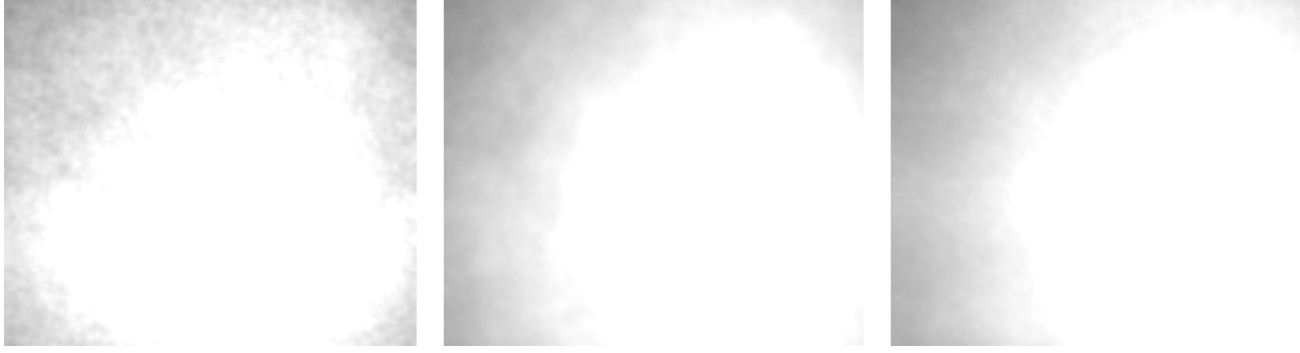

*Figure 5.* Example Input of Registration (ffp illumination profiles)

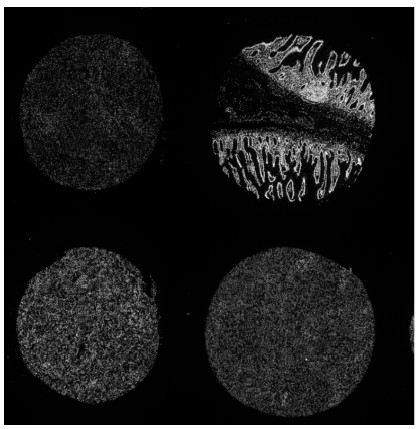 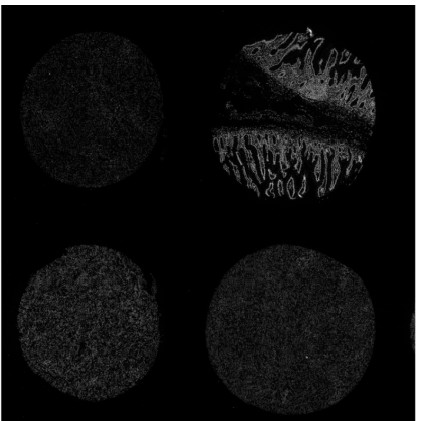 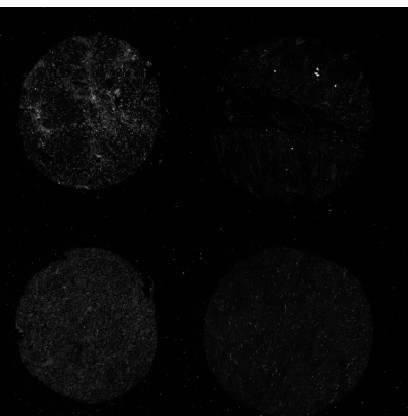

*Figure 6.* Example Input of Background Subtraction/Dearray (stitched OME-TIFF)

## D. Cell Annotation Challenge: Evaluation Details

This appendix provides implementation details for the Cell Annotation Challenge evaluation pipeline.

### D.1. Input Format and Data Structure

The evaluation framework operates on two CSV files with distinct structural requirements:

- **Agent Output (AI CSV):** Must contain at least three columns:
  - `cell_id`: Unique identifier for each cell.
  - `cluster`: Cluster assignment from unsupervised clustering (inherited from input).
  - `Annotation`: Agent-assigned cell type label.

- **Ground Truth (GT CSV):** Must contain at least two columns:
  - `cell_id`: Matching cell identifiers.
  - `Annotation`: Ground Truth cell type label.

### D.2. Cell Matching and Cluster Aggregation

Agents are explicitly instructed that all cells within the same input cluster must be annotated with an identical label. Therefore, the evaluation matches annotations strictly at the cluster level rather than assessing individual cells.

### D.3. Semantic Similarity Computation

Rather than naive string matching (which would fail on synonyms like "T helper cell" vs. "CD4+ T cell"), we employ semantic similarity using the CyteOnto (Ahuja et al., 2025) framework:

**Step 1: Text Embedding.** Labels from both the agent and the Ground Truth are first expanded into comprehensive biological descriptions using an Claude Sonnet 4 before embedding. Descriptions follow the official CyteOnto implementation (Ahuja et al., 2025). All unique cell type descriptions are then encoded using the Qwen3-Embedding-8B model, producing 4096-dimensional vector representations.

**Step 2: Cosine Similarity.** For each cluster, we compute the cosine similarity between the agent label description embedding $\mathbf{e}_{ai}$ and ground truth label description embedding $\mathbf{e}_{gt}$:

$$\cos(\mathbf{e}_{ai}, \mathbf{e}_{gt}) = \frac{\mathbf{e}_{ai} \cdot \mathbf{e}_{gt}}{\|\mathbf{e}_{ai}\| \|\mathbf{e}_{gt}\|} \tag{1}$$

**Step 3: Gaussian Heat Kernel (GHK) Transform.** To penalize semantic drift more aggressively, the raw cosine similarity $S_{\text{cos}}$ is transformed via a Gaussian kernel centered at 1.0:

$$\text{GHK}(\mathbf{e}_{\text{ai}}, \mathbf{e}_{\text{gt}}) = \exp\left(-\frac{(S_{\text{cos}}(\mathbf{e}_{\text{ai}}, \mathbf{e}_{\text{gt}}) - 1)^2}{2\sigma^2}\right) \tag{2}$$

where $S_{\text{cos}}$ is the cosine similarity calculated in Step 2 and $\sigma = 0.25$ controls the bandwidth. This transformation maps the similarity to the $[0, 1]$ range, creating a steeper penalty for deviations from exact matches compared to the linear cosine scale.

### D.4. Consistency Validation

Before evaluation proceeds, the pipeline validates that all cells within each cluster have identical agent-assigned annotations. Submissions with intra-cluster annotation inconsistencies are rejected with detailed error reports specifying which clusters contain conflicting labels and their respective counts.

## E. CRC-CODEX Quantification Challenge Details

### E.1. Dataset Description

The CRC-CODEX Quantification Challenge uses 10 high-resolution tissue microarray (TMA) core images from the Colorectal Cancer CODEX dataset (Schürch et al., 2020). Each image contains multiplexed protein marker measurements across thousands of cells. The ground truth quantification was obtained from the original human-verified dataset.

### E.2. Evaluation Protocol

Each agent was provided with the raw OME-TIFF image file and a markers CSV file specifying the channel-to-marker mapping. Agents were instructed to perform end-to-end quantification, involving:

1. Nuclear/cell segmentation.

2. Single-cell feature extraction.

Each agent quantified all 10 images for 3 trials to assess consistency. Reported values are mean and standard deviations across all trials.

### E.3. Metric Definitions

All metrics are designed to be segmentation-tolerant: they evaluate whether the agent captures the correct biological signal without requiring exact cell-to-cell correspondence (ID matching) with the ground truth segmentation.

#### E.3.1. MARKER ABUNDANCE CORRELATIONS

Pearson $r$ and Spearman $\rho$ are computed on **marker sums** (total intensity per marker across all cells) to capture the total signal abundance:

$$\text{MarkerSum}_m = \sum_{i=1}^{N} I_{i,m} \tag{3}$$

where $I_{i,m}$ is the intensity of marker $m$ in cell $i$.

- **Pearson $r$:** Measures linear correlation between agent and GT marker abundances.

- **Spearman $\rho$:** Measures rank correlation (preserving relative high/low marker hierarchy).

Both metrics range from -1 to 1, with 1 indicating perfect agreement.

### E.3.2. CORRELATION MATRIX SIMILARITY

This metric measures the preservation of marker-marker co-expression patterns (e.g., ensuring CD4 and CD3 correlate).

1. Apply arcsinh transformation (cofactor = 5) to all marker intensities.

2. Compute the Spearman correlation matrix for each dataset (Agent, GT).

3. Extract the upper triangle (excluding the diagonal) to avoid inflation from self-correlations.

4. Compute the cosine similarity between the flattened upper triangles:

$$\text{CosSim} = \frac{\mathbf{u}_{\text{agent}} \cdot \mathbf{u}_{\text{GT}}}{\|\mathbf{u}_{\text{agent}}\| \cdot \|\mathbf{u}_{\text{GT}}\|} \tag{4}$$

### E.3.3. MAXIMUM MEAN DISCREPANCY (MMD)

MMD measures the distance between multivariate phenotype distributions using a kernel embedding approach (Gretton et al., 2012):

$$\text{MMD}^2 = \mathbb{E}[k(X, X')] + \mathbb{E}[k(Y, Y')] - 2\mathbb{E}[k(X, Y)] \tag{5}$$

where $k$ is a Radial Basis Function (RBF) kernel:

$$k(x, y) = \exp\left(-\gamma \|x - y\|^2\right) \tag{6}$$

**Implementation details:**

- **Bandwidth selection:** Median heuristic ($\gamma = 1/(2 \cdot \text{median\_dist}^2)$) computed on a random subsample of 1,000 cells.

- **Input transformation:** All marker intensities are arcsinh-transformed ($cf = 5$) before computing phenotype vectors.

- **Estimator:** Unbiased U-statistic (diagonal terms excluded).

- **Random seed:** Fixed at 42 for reproducibility.

Lower MMD indicates more similar cell population distributions.

### E.3.4. GRID-BASED SPATIAL SPEARMAN CORRELATION

This metric verifies that biological signal is localized correctly in space without requiring cell-to-cell coordinate matching:

1. **Grid construction:** Overlay a uniform grid (100 pixels) on the tissue image.

2. **Tile aggregation:** For each tile, compute the **mean** intensity per marker. Using the mean (rather than sum) normalizes by cell count, making the metric robust to local differences in segmentation density.

3. **Correlation:** Compute Spearman correlation between Agent and GT tile intensities for each marker.

4. **Aggregation:** Report the median correlation across all valid markers.

$$\rho_{\text{spatial}} = \text{median}_m \left[\text{Spearman}(\bar{I}_{t,m}^{\text{agent}}, \bar{I}_{t,m}^{\text{GT}})\right] \tag{7}$$

where $\bar{I}_{t,m}$ is the mean intensity of marker $m$ in tile $t$.

