# OpenReview forum: "SP-Mind: An Autonomous Reasoning Agent for Spatial Proteomics Analysis"
_ICML.cc/2026/Conference — ICML 2026 regular_

### Official Review · Reviewer_yPqZ · 2026-03-10

**Soundness:** 3
**Presentation:** 3
**Significance:** 3
**Originality:** 2
**Overall Recommendation:** 3
**Confidence:** 3

**Summary:**

The paper introduces an AI agent designed for spatial proteomics analysis from scratch. Using natural language queries, their method SP-Mind, automatically executes a series of tasks without any fine-tuned method. SP-Mind is based on ReAct-style  loop with prior knowledge, with Claude-Sonnet models as the base models, and uses code-based environment to execute tasks. They also introduce SP-Bench to measure performance of agents on such analysis. Overall, they show that SP-Mind almost always outperforms existing methods on multiple benchmarks related to multiplex imaging.

**Compliance With Llm Reviewing Policy:**

Affirmed.

**Final Justification:**

I stick with my weak reject claim. While the direction of the work is interesting, some questions about novelty do arise. Moreover, I am not sure if ICML is the correct venue for this task, given the ease of task it might provide to wet-lab/computational biologists.

**Key Questions For Authors:**

Mentioned along with weaknesses.

**Limitations:**

Mentioned along with weaknesses.

**Strengths And Weaknesses:**

Strengths:
- Task automation for SP analysis is quite needed. Since SP pipelines often contain lot of steps, and often manual intervention, if we can automate some steps, it will make the pre-processing much faster and cheaper. Other general-purpose agents for bio can also do these tasks, but none other than this are specialized or have tools to execute those
- I like the integration of comp bio softwares here than just functions. This is a good start, since a lot of softwares are mainly designed to be UI-based & not API-based, and require manual effort.
- Given the lack of evaluation methods, releasing SP-Bench is good for the community, as it contains lot of questions at different levels of effort. I hope the benchmark is released on acceptance, and continuously updated (since the size is quite small).

Weaknesses:
- From a novelty perspective, the contributions in the agent are slightly limited. While having these expert curated templates are nice to have, ReAct/CodeAct agents ([1]) and tool libraries ([2,3]) already exist and have been explored.
- I feel one driving factor for the method is these skill templates, however no examples are provided. It would've been nice to see them + example conversations of SP-Mind and baselines to see the difference. Moreover, how are these templates collected? how many experts were involved? how many templates are created? What is the fleiss kappa?
- Recently, it has been seen that evaluation on a lot of medical benchmarks result in extremely variable results from the agents when run multiple times. Most of the runs here have just the mean reported (or a single run). Is it possible to run the benchmark multiple times to get confidence intervals and statistical significance?
- While the questions and benchmark state general SP analysis, aren't there differences in analysis of IMC and CODEX? Does the tools take care of these?

Overall, I think the aim of the paper is strong and beneficial to the bio / SP community. However, I feel the methodological novelty is limited. I am looking forward to having a discussion with the authors but I lean to weak reject.

[1] Huang, Kexin, et al. "Biomni: A general-purpose biomedical ai agent." _biorxiv_ (2025).

[2] Gao, Shanghua, et al. "Democratizing AI scientists using ToolUniverse." _arXiv preprint arXiv:2509.23426_ (2025).

[3] Sui, Pengwei, et al. "Medea: An omics AI agent for therapeutic discovery." _bioRxiv_ (2026): 2026-01.

---

> ### Author Rebuttal · Authors · 2026-03-31
>
> We thank the reviewer for the thoughtful assessment of our work. We now address their concerns.
>
> **W1.** *Novelty*
>
> **A.** We thank the reviewer for this observation. We would like to clarify the core contribution of this paper: (1) systematically defines and formalizes the problem of automated spatial proteomics analysis **for the first time**; (2) establishes SP-Bench, the **first standardized benchmark** in this domain; and (3) provides a **complete and validated solution**, SP-Mind, to address this problem. These three parts are intended as a unified contribution.
>
> Importantly, the design of SP-Mind is not straightforward. Making autonomous reasoning work in spatial proteomics requires substantial domain-specific design: decomposing the analysis workflow into **effective Spatial BioSkill Templates**; integrating **real-world toolchains** such as MCMICRO; and the error-handling mechanisms are **specifically designed for failure modes common in computational biology workflows**. These challenges cannot be resolved by generic ReAct-style prompting alone. The experiment results support this point. **Generic or less specialized agents perform substantially worse on the same tasks.**
>
> **W2.** *Skill templates example & details.*
>
> **A.** Thanks for the feedback. In the revision, we will add concrete template examples, the full construction process. A condensed example of a skill template (annotation) is:
>
> - **Instruction:** Perform spatial proteomics cell type annotation for clustered cells.
> - **Reasoning policy:** Use **rank-based analysis** rather than relying only on absolute expression values. For each marker, compare its ranking across clusters.
> - **Decision rule:** Infer labels from **marker combination patterns**, not single markers alone.
> - **Cluster-specific evidence:** For each cluster, identify the markers that are most **unique relative to other clusters**, since these provide the strongest evidence for annotation.
> - **Constraint handling:** Account for **panel limitations** and missing canonical markers. When key markers are unavailable, use alternative lineage markers where appropriate.
> - **Execution/output requirements:** save both the full annotated output and a concise `annotation_summary.csv` containing `Cluster, Annotation` pairs for verification.
>
> Regarding collection, the templates were **collaboratively designed by 3 domain experts** through 4 rounds of iterative refinement, grounded in established spatial proteomics workflows and repeatedly validated on real tasks (150+). Because this was a rigorous **collaborative design process rather than independent annotation**, inter-annotator agreement metrics such as **Fleiss’ kappa are not directly applicable**.
>
> We also note that the current paper already includes a qualitative comparison for the cell-annotation setting (Fig. 3), where SP-Mind’s behavior is contrasted against a baseline agent without such skills. We plan to include more in the revised version.
>
> **W3.** *Running the benchmark multiple times*
>
> **A.** We conducted new independent repeated-run experiments and summarize the results in Tables 1–3. For consistency and standardization, we repeated each evaluation suite **3 times**, and we will incorporate these additional results into the revised manuscript. Across SP-Bench, CRC-CODEX quantification Challenge, and Cell Annotation Challenge, SP-Mind shows both the strongest overall performance and competitive stability across independent runs. SP-Mind achieves an aggregate score of **68.9** on **SP-Bench** (Table 1) and **0.6811** on **Cell Annotation** (Table 3), and similarly, strongest results on CRC CODEX metrics.
>
> Here is the link to the result table (including std):
>
> https://i.postimg.cc/TYNdHKTB/table-1.png
>
> **W4.** *Differences in IMC and CODEX*
>
> **A.** Yes. IMC and CODEX differ substantially, and we agree that this should be stated more clearly. SP-Mind is designed to handle **modality-specific execution** through tool selection: for fluorescence-based imaging such as CODEX/CyCIF, the agent can compose modular tools for the workflow; for IMC, we also integrated the **TRACERx-PHLEX** [1] pipeline as a callable option for the agent.
>
> However, **SP-Bench was intentionally designed around granular, modular tasks** that can be independently evaluated and then composed into progressively more complex workflows. In contrast, **TRACERx-PHLEX is primarily an end-to-end IMC pipeline with tightly coupled internal stages**. As a result, it is useful as an execution backend, but it is **not well matched to the stage-level evaluation design of SP-Bench**. We will clarify this distinction in the revision. We agree that modality coverage is important, and **future versions of SP-Bench will aim to incorporate more modular IMC-specific tasks** to better reflect these analysis differences.
>
> [1] Magness, Alastair, et al. "Deep cell phenotyping and spatial analysis of multiplexed imaging with TRACERx-PHLEX." Nature Communications 15 (2024): 5135.

---

> > ### Author Rebuttal · Reviewer_yPqZ · 2026-04-04
> >
> > Thanks for the rebuttal. Some of my concerns are alleviated.
> >
> > While the authors say that "systematically defines and formalizes the problem of automated spatial proteomics analysis for the first time", I feel the workflow is just how one does standard analysis. I agree that they aim to automate it with the agents but the workflow isn't the novel aspect.
> >
> > Can the authors share one of the skill templates here? I understand that the rebuttal had a short character limit.

---

> > > ### Author Response · Authors · 2026-04-04
> > >
> > > We thank the reviewer for the helpful follow-up. We agree that the biological workflow itself is not the novelty—the core contribution is not proposing a new spatial proteomics pipeline, but rather **systematically formulating autonomous spatial proteomics analysis as an agentic systems problem**, and providing the first integrated benchmark-and-solution framework for it. In this work, the novelty lies in: **(1)** formalizing how natural-language spatial proteomics requests can be mapped into executable multi-stage workflows, **(2)** establishing **SP-Bench** as the first standardized benchmark for this setting, and **(3)** developing **SP-Mind** as a validated domain-specific solution.
> > >
> > > Per reviewer’s request, below we provide the cell annotation skill template to make the design more explicit:
> > >
> > > ```text
> > > # Cell Type Annotation Skill
> > >
> > > ## Analysis Methodology
> > >
> > > ### 1. Use Rank-Based Analysis
> > > - Do NOT rely solely on absolute expression values per cluster
> > > - For each marker, calculate its **ranking** across all clusters
> > > - Focus on which clusters rank highest for specific markers
> > > - Even with low absolute values, the cluster ranking #1 for a marker may still represent that cell type
> > >
> > > ### 2. Consider Marker Combination Patterns
> > > - Do NOT make annotations based on a single highly-expressed marker
> > > - Consult your knowledge of classic marker combinations for each cell type
> > > - Consider both **positive markers** AND **negative markers** (exclusion markers)
> > >
> > > ### 3. Identify Each Cluster's Unique Markers
> > > - For each cluster, find markers that are **most unique relative to other clusters**
> > > - Identify which markers rank #1 or #2 in that cluster
> > > - Look for markers where a cluster is significantly higher than all others
> > > - These unique markers are the strongest evidence for cell type identity
> > >
> > > ### 4. Be Aware of Panel Limitations
> > > - The marker panel may lack classic markers for certain cell types
> > > - If a key marker is missing, use alternative lineage markers
> > > - For clusters that are difficult to clearly determine, use generic labels like "Other" or "Unknown"
> > >
> > > ### 5. Workflow Summary
> > > 1. Load data and calculate mean expression per cluster
> > > 2. Calculate rankings: for each marker, rank clusters from highest to lowest
> > > 3. For each cluster, identify its top-ranked markers
> > > 4. Match marker combinations to known cell type signatures
> > > 5. Verify with negative markers (what the cluster does NOT express)
> > > 6. Assign annotations based on the combined evidence
> > >
> > > ## Output Requirements
> > >
> > > In addition to the full annotated CSV file, you MUST also save a summary file called `annotation_summary.csv` in the same output directory, which should contain:
> > >
> > > - **Cluster**: The cluster ID (integer)
> > > - **Annotation**: The assigned cell type label
> > >
> > > This makes it easy to quickly verify the annotation results.
> > > ```
> > >
> > > A template in SP-Mind is not just a label or short prompt; it provides **task-specific expert guidance** on how the agent should reason, what evidence it should prioritize, how it should handle common failure modes or ambiguity, and what outputs it must produce for downstream compatibility.
> > >
> > > We would be happy to address any further questions.

---

### Official Review · Reviewer_FwRS · 2026-03-10

**Soundness:** 3
**Presentation:** 3
**Significance:** 3
**Originality:** 2
**Overall Recommendation:** 4
**Confidence:** 3

**Summary:**

This paper introduces SP-Mind, an agent framework capable of tackling multiple tasks in spatial proteomics analysis. Specifically, it takes user requests as inputs, selects and calls different task-specific tools, and accomplishes the given tasks with corresponding outputs. In addition, the paper introduces a new evaluation benchmark, SP-Bench, which consists of 102 distinct spatial proteomics queries spanning 18 categories and 8 analysis stages. The experimental results demonstrate that SP-Mind achieves state-of-the-art performance on SP-Bench, the CRC-CODEX challenge, and the cell annotation challenge.

**Compliance With Llm Reviewing Policy:**

Affirmed.

**Final Justification:**

The authors have addressed my concerns. Although I consider the work to lack technical novelty, it represents a solid engineering effort and makes a valuable contribution to the community, as it automates the entire analysis pipeline.

**Key Questions For Authors:**

1. Could you provide the prompts that you used for different stages in SP-Mind?

**Limitations:**

yes

**Strengths And Weaknesses:**

**Strengths**

1. The motivation is convincing and the problem setting is significant.
2. The experimental comparisons are comprehensive.
3. The proposed method is well-designed and sound.

**Weaknesses**

1. The technical novelty is limited. The work appears to be more of an engineering contribution that integrates multiple analytic tools via an agent framework (ReAct).
2. Ablation studies on SP-Mind are lacking. For instance, the authors could provide experimental results ablating the thought process to better justify this design choice.
3. It would be beneficial to present some failure cases. In Supplementary B.8, the authors introduce several mechanisms to handle errors; experimental results evaluating the effectiveness of each design would strengthen the validity of these choices.

---

> ### Author Rebuttal · Authors · 2026-03-31
>
> We thank the reviewer for recognizing that the motivation is convincing and that the experimental comparisons are comprehensive. We now address their concerns.
>
> **W1.** *“The technical novelty is limited…”*
>
> **A.** We thank the reviewer for this observation. We would like to clarify that the core contribution of this work is not simply applying ReAct to a new setting, but introducing an integrated foundation for autonomous spatial proteomics analysis. Specifically, this paper: (1) systematically defines and formalizes the problem of automated spatial proteomics analysis **for the first time**; (2) establishes SP-Bench, the **first standardized benchmark** in this domain; and (3) provides a **complete and validated solution**, SP-Mind, to address this problem. These three parts are intended as a unified contribution.
>
> Importantly, the design of SP-Mind is not straightforward. Making autonomous reasoning work in spatial proteomics requires substantial domain-specific design: decomposing the analysis workflow into **effective Spatial BioSkill Templates**, integrating **real-world toolchains** such as MCMICRO, and the error-handling mechanisms are **specifically designed for failure modes common in computational biology workflows**. These challenges cannot be resolved by generic ReAct-style prompting alone. The experiment results support this point. **Generic or less specialized agents perform substantially worse on the same tasks.** This gap indicates that the contribution is not merely incremental, but comes from the necessity and effectiveness of the proposed domain-specific system design.
>
> **W2.** *“Ablation studies on SP-Mind are lacking…”*
>
> **A.** To directly address the reviewer’s request, we conducted new **ablation experiments** designed to isolate the contribution of the **Spatial BioSkill Templates** from that of SP-Mind architecture itself. For consistency and standardization, we repeated each evaluation suite **3 times**, and we will incorporate these additional results into the revised manuscript. The results show that the specialized tool suite alone is not sufficient to recover the full performance of SP-Mind, particularly when the task gets complicated (e.g. chaining all 3+ basic stages together). On **SP-Bench**, the aggregate score drops from **68.9** to **62.4** (**-6.5 points**) without skills. This reduction is especially pronounced in the **Advanced** tier, and also in the **Challenging** tier. Interestingly, in the Cell Annotation Challenge, adding skill templates provides the clearest benefits on cHL datasets. We do observe one exception on **PDAC IMC**, where the no-skill variant performs better, indicating that the benefit of expert-guided reasoning is not uniform across all datasets.
>
> Here is the link to the result table:
>
> https://i.postimg.cc/TYNdHKTB/table-1.png
>
> **W3.** *Failure Cases & Error Recovery*
>
> **A.** We thank the reviewer for this helpful suggestion. We will include representative failure cases in the revision. A representative failure mode is **file-management / state-tracking error**: after successfully completing several stages, the agent may place an intermediate result into the wrong folder or directory structure. Frequency ~15%. A second failure mode arises in **background subtraction**, where it requires additional input engineering (e.g., adapting `markers.csv` into compatible format). SP-Mind can usually handle this when background subtraction is an isolated task, but in longer chained workflows it may fail to sufficiently inspect the error (frequency ~40%).
>
> The recovery mechanisms in Supplementary B.8’s roles are primarily to improve robustness under realistic execution settings. **Transparent Runtime Fallback** is important when the same workflow is executed across different environments (e.g., HPC versus local macOS). **Observation-Driven Recovery** is a major contributor to success in practice when the initial tool invocation does not succeed. **Execution-Level Signaling** serves as a guardrail to help the agent differentiate zero results vs real runtime failures (which aids the agent in producing summary for the client).
>
> **Q1.** *Prompts at Different Stages*
>
> **A.** Thank you for raising this point. SP-Mind does not use a separate fully handcrafted prompt for each benchmark stage. Instead, it uses a **modular prompting scheme** composed of: (1) a fixed **base system prompt** defining the agent role, execution environment, available tools, and data-first reasoning protocol; (2) standardized **tool descriptions and usage guidance**; and (3) a dynamically injected **Spatial BioSkill Template** selected based on the user query/task type. Thus, the effective prompt is: **[Base prompt] + [tool specifications] + [task-relevant skill template] + [user query]**. Because of the rebuttal length limit, we cannot include the full exact prompts here. We will include them in revision.

---

> > ### Author Rebuttal · Reviewer_FwRS · 2026-04-02
> >
> > I appreciate the responses and they have addressed my concerns.

---

> > > ### Author Response · Authors · 2026-04-03
> > >
> > > We sincerely appreciate your recognition that our rebuttal has adequately addressed the concerns regarding technical novelty, ablation analysis, failure cases, and prompting details.
> > >
> > > Given that you now consider these concerns to be fully resolved, we would be very grateful if you would consider updating the overall score to reflect this assessment. In particular, we believe that the new ablation results, repeated-run analysis, and clarified failure-case discussion substantially strengthen the paper and address the main concerns underlying the initial weak reject recommendation.
> > >
> > > Thank you again for your thoughtful review and for the time you invested in helping improve the paper. We would be happy to address any further questions.

---

### Official Review · Reviewer_6jnU · 2026-03-11

**Soundness:** 3
**Presentation:** 3
**Significance:** 2
**Originality:** 3
**Overall Recommendation:** 5
**Confidence:** 4

**Summary:**

This paper presents SP-Mind, an LLM-based autonomous agent for end-to-end spatial proteomics analysis. The system combines a ReAct and CodeAct style reasoning loop, a modular library of spatial proteomics tools spanning preprocessing through clustering, and expert-curated “Spatial BioSkill Templates” that guide tool selection, parameter choice, and error recovery. To evaluate the system, the authors introduce SP-Bench, a benchmark of 102 natural-language tasks across 18 categories and 4 difficulty tiers covering 8 stages of the spatial proteomics workflow. In addition to SP-Bench, the paper evaluates downstream performance on a CRC-CODEX quantification challenge and a large-scale cell annotation challenge.

**Compliance With Llm Reviewing Policy:**

Affirmed.

**Key Questions For Authors:**

1. Please clarify the exact adjudication protocol for SP-Bench execution accuracy. How is “fulfills the query objective” operationalized across tasks with multiple valid workflows, parameter choices, or output formats?

2. Since SP-Bench queries were initially generated with an LLM and then expert-validated, can the authors provide more detail on benchmark construction quality control, inter-expert agreement, and safeguards against style or distribution bias that may favor their own prompting strategy?

3. Can you provide stronger ablations isolating the contribution of the BioSkill Templates from the contribution of the specialized tool suite itself?

**Limitations:**

yes

**Strengths And Weaknesses:**

Strengths

1. The paper targets an important and underexplored application area. Autonomous analysis for spatial proteomics is practically meaningful, and the work goes beyond toy agent tasks to address a real scientific workflow.

2. The benchmark contribution is potentially valuable. SP-Bench covers many stages of the spatial proteomics pipeline and could become a useful evaluation resource for future domain-specific agent research.

3. The empirical results are promising. SP-Mind shows consistent gains over the reported baselines on the new benchmark and on downstream tasks, suggesting that domain-specific skills and tools can materially improve agent performance.

Weaknesses

1. The main benchmark evaluation is not rigorous enough. The definition of execution success is not sufficiently transparent for complex scientific workflows where multiple valid outputs and parameter choices may exist.

2. The benchmark may contain construction bias. Since SP-Bench queries were LLM-generated and then expert-validated, it is unclear whether the benchmark distribution may favor the proposed system’s prompting and reasoning style.

---

> ### Author Rebuttal · Authors · 2026-03-31
>
> We thank the reviewer for their thoughtful and positive assessment of our submission, and for recognizing the potential value of SP-Bench as a community resource, and the promising empirical gains demonstrated by SP-Mind. We now address their concerns.
>
> **W1 & Q1.** *“The main benchmark evaluation is not rigorous enough... How is ‘fulfills the query objective’ operationalized across tasks with multiple valid workflows, parameter choices, or output formats?”*
>
> **A.** We agree that evaluation transparency is especially important for scientific workflows. We would like to clarify that, as defined in Appendix C.3, SP-Bench execution success is **not** determined by mere code completion, sequential tool invocation, or file generation. Rather, a query is counted as successful only when the agent completes the requested workflow **and** satisfies an **Output Fidelity** criterion: the produced artifact must be biologically valid and appropriate for the intended analytical objective, including suitability for downstream stages when relevant. This criterion was manually assessed by **three bioinformatics experts**. Importantly, SP-Bench does **not** assume a single canonical workflow, parameterization, or output format. If a query explicitly specifies a required configuration or parameter, the agent is expected to follow it. Otherwise, experts evaluated whether the selected tools and parameter choices were biologically reasonable and whether the resulting output fulfilled the query objective. We will revise the paper to make this adjudication protocol more explicit.
>
> **W2 & Q2.** *“...more detail on benchmark construction quality control, inter-expert agreement, and safeguards against style or distribution bias that may favor their own prompting strategy?”*
>
> **A.** We thank the reviewer for raising this important point. In our case, however, the query-generation model and the evaluated agent are **intentionally separated**: SP-Bench queries were generated using Gemini 3 Pro, whereas SP-Mind and all baselines were run with Claude Sonnet 4 as the backbone, with no shared chat history, prompt context, or interaction traces. Moreover, these models exhibit different reasoning and response styles, which reduces the likelihood that benchmark performance is driven by stylistic alignment to our agent.
>
> More importantly, benchmark construction included substantial expert quality control. 3 bioinformatics experts checked (1) biological plausibility, (2) correct workflow/module dependencies, and (3) alignment with realistic user requests in spatial proteomics practice. The experts did not merely filter queries; they also **revised and refined** them when needed to ensure quality and realism. Queries judged problematic by **at least two of the three experts** (32 out of 134) were excluded.  It is also important to note that these three experts had **no access to the design details or prompting strategies** of SP-Mind or any baseline agent during benchmark construction, reducing the risk that query selection was unintentionally aligned to a particular system. We will clarify these safeguards more explicitly in the revision.
>
> **Q3.** *“...ablations isolating the contribution of the BioSkill Templates”*
>
> **A.** To directly address the reviewer’s request, we conducted new **ablation experiments** designed to isolate the contribution of the **Spatial BioSkill Templates** from that of SP-Mind architecture itself. In addition, we note that **Biomni-SP** and **ToolUniverse-SP**, which are equipped with the same specialized spatial proteomics tool suite but use different agent frameworks, already provide a complementary point of comparison for the value of the tool suite itself. For consistency and standardization, we repeated each evaluation suite **3 times**, and we will incorporate these additional results into the revised manuscript.
>
> The results show that the specialized tool suite alone is not sufficient to recover the full performance of SP-Mind, particularly when the task gets complicated (e.g. chaining all 3+ basic stages together, like stitching + background subtraction + dearray, demands careful call of each tools, accurate storage of intermediate results, and potentially doing further processing of outputs before feeding into the next downstream stage). On **SP-Bench**, the aggregate score drops from **68.9** to **62.4** (**-6.5 points**) without skills. This reduction is especially pronounced in the **Advanced** tier, and also in the **Challenging** tier. Interestingly, in the Cell Annotation Challenge, adding skill templates provides the clearest benefits on cHL datasets. We do observe one exception on **PDAC IMC**, where the no-skill variant performs better, indicating that the benefit of expert-guided reasoning is not uniform across all datasets.
>
> Here is the link to the result table:
>
> https://i.postimg.cc/TYNdHKTB/table-1.png

---

> > ### Author Rebuttal · Reviewer_6jnU · 2026-04-05
> >
> > The authors have addressed my comments.

---

> > > ### Author Response · Authors · 2026-04-05
> > >
> > > We appreciate the reviewer’s recognition that this work addresses an **important and underexplored application area**, that **SP-Bench can serve as a valuable resource for future domain-specific agent research**, and that **SP-Mind demonstrates promising and consistent empirical gains over baselines** on both the benchmark and downstream tasks. We are also grateful that the reviewer found our rebuttal to have **adequately addressed all concerns**. Thank you!

---

### Official Review · Reviewer_igYA · 2026-03-13

**Soundness:** 3
**Presentation:** 3
**Significance:** 3
**Originality:** 2
**Overall Recommendation:** 5
**Confidence:** 5

**Summary:**

This paper presents SP-Mind, an LLM-driven autonomous reasoning agent designed to orchestrate end-to-end spatial proteomics analysis workflows. Spatial proteomics analysis typically requires a complex pipeline involving multiple stages such as illumination correction, image registration, cell segmentation, quantification, and downstream spatial analysis. The authors propose an agent framework that converts natural-language user queries into executable analysis workflows by combining a ReAct-style reasoning loop, a modular spatial proteomics tool library, and domain-specific skill templates encoding expert workflow knowledge. To evaluate the proposed system, the authors introduce SP-Bench, a benchmark consisting of 102 natural language tasks covering eight major stages of spatial proteomics pipelines with different difficulty levels. Experiments compare SP-Mind against several agent baselines, including AutoGen, Biomni, and ToolUniverse (and their spatially adapted variants). Results show that SP-Mind achieves higher execution success rates on SP-Bench and demonstrates improved performance on downstream quantification and cell annotation tasks. Overall, the paper aims to demonstrate that domain-aware LLM agents can effectively orchestrate complex biological analysis workflows and potentially reduce the need for manual pipeline configuration.

**Compliance With Llm Reviewing Policy:**

Affirmed.

**Final Justification:**

This paper studies the problem of autonomous orchestration of spatial proteomics workflows using LLM-based agents, and introduces both a system (SP-Mind) and a benchmark (SP-Bench) for this purpose.

In terms of strengths, the work is well-motivated and addresses an important and practical problem. The system design is clearly presented and integrates multiple components (reasoning, tool use, and domain-specific skill templates) in a coherent manner. The introduction of SP-Bench is particularly valuable, as it provides a structured benchmark for evaluating agent-based scientific workflows, which is currently lacking in this domain. The experimental results are comprehensive and demonstrate consistent improvements over relevant baselines.

In terms of weaknesses, my main concern remains the level of technical novelty. The core methodology largely builds upon existing agent paradigms (e.g., ReAct-style reasoning and tool-based execution), and the contribution is primarily at the level of system integration and domain adaptation rather than fundamentally new algorithmic or modeling advances. In addition, while the paper includes some evaluation of output quality, the assessment of biological validity could be further strengthened with more explicit and standardized quantitative metrics.

The rebuttal partially addressed my concerns. The authors provided additional repeated-run experiments that help clarify the stability of the system, and the added failure case analysis improves the transparency of the method. The clarification on output fidelity and expert evaluation is also helpful. However, these additions do not fundamentally change my assessment regarding the level of methodological novelty and the limitations of the evaluation.

**Key Questions For Authors:**

1. LLM-based agents can exhibit stochastic behavior across runs. Could the authors provide additional analysis on the variance of SP-Mind’s performance across multiple independent runs?

2. Beyond execution success, have the authors evaluated the biological validity of the generated outputs (e.g., segmentation quality, clustering accuracy, or phenotype annotation consistency)?

3. Failure case analysis. Could the authors provide examples of cases where the agent generates incorrect workflows or fails to complete a task, and discuss potential reasons for these failures?

**Limitations:**

yes.

**Strengths And Weaknesses:**

Strengths
1. Important and well-motivated application problem. Spatial proteomics analysis pipelines are complex and typically require significant manual orchestration across multiple specialized tools. Automating such workflows using LLM-based agents is a meaningful and timely direction that could reduce the barrier to entry for spatial biology analysis.
2. The proposed SP-Mind framework integrates several complementary components, including a reasoning agent, a curated spatial proteomics tool library, and domain-specific skill templates. The overall architecture is clearly described and provides a logical approach for converting natural language queries into executable scientific workflows.
3. The use of skill templates to encode expert workflow knowledge is a practical way to guide the agent in selecting tools and configuring parameters. This design helps constrain the reasoning process and reduces the likelihood of generating invalid analysis pipelines.
4. The proposed SP-Bench benchmark provides a structured evaluation framework for spatial proteomics analysis tasks with different levels of workflow complexity. Such a benchmark could be valuable for future research on LLM-based scientific agents.

Weaknesses
1. While the system is well engineered, the core approach mainly builds upon existing agent frameworks (e.g., ReAct-style reasoning and tool-based execution) combined with domain-specific prompt templates. The technical contribution therefore appears somewhat incremental.
2. The main benchmark metric focuses on whether the agent successfully executes a workflow, but provides limited evaluation of the biological validity or quality of the resulting analyses.
3. Limited analysis of reasoning reliability. LLM-based agents can exhibit hallucination or unstable reasoning behaviors, yet the paper does not provide a detailed analysis of reasoning errors or performance variance across multiple runs.

---

> ### Author Rebuttal · Authors · 2026-03-31
>
> We thank the reviewer for their thoughtful and positive assessment of our submission and for highlighting the *“important and well-motivated application problem”*, the clarity of the SP-Mind framework, and the potential value of SP-Bench to future research. We now address their concerns.
>
> **W1.** *“…The technical contribution therefore appears somewhat incremental.”*
>
> **A.** We thank the reviewer for this observation. We would like to clarify that the core contribution of this work is not simply applying ReAct to a new setting, but introducing an integrated foundation for autonomous spatial proteomics analysis. Specifically, this paper: (1) systematically defines and formalizes the problem of automated spatial proteomics analysis **for the first time**; (2) establishes SP-Bench, the **first standardized benchmark** in this domain; and (3) provides a **complete and validated solution**, SP-Mind, to address this problem. These three parts are intended as a unified contribution.
>
> Importantly, the design of SP-Mind is not straightforward. Making autonomous reasoning work in spatial proteomics requires substantial **domain-specific design**: decomposing the analysis workflow into effective Spatial BioSkill Templates **demands deep expert knowledge**; integrating real-world toolchains such as MCMICRO [1] requires **nontrivial adaptation of complex bioinformatics pipelines into an agent-compatible modular system**; and the error-handling mechanisms are **specifically designed for failure modes common in computational biology workflows**. These challenges cannot be resolved by generic ReAct-style prompting alone. The experiment results support this point. **Generic or less specialized agents (e.g., Biomni-SP) perform substantially worse on the same tasks.** This gap indicates that the contribution is not merely incremental, but comes from the necessity and effectiveness of the proposed domain-specific system design.
>
> [1] Schapiro, Denis, et al. "MCMICRO: a scalable, modular image-processing pipeline for multiplexed tissue imaging." Nature Methods 19 (2022): 311–315.
>
> **Q1 & W3.** *”...Could the authors provide additional analysis on the variance of SP-Mind’s performance across multiple independent runs?”*
>
> **A.** To address the reviewer’s concerns on LLM instability and run-to-run variance, we conducted new independent repeated-run experiments and summarize the results in Tables 1–3. For consistency and standardization, we repeated each evaluation suite **3 times**, and we will incorporate these additional results into the revised manuscript. Across SP-Bench, CRC-CODEX quantification Challenge, and Cell Annotation Challenge, SP-Mind shows both the strongest overall performance and competitive stability across independent runs. SP-Mind achieves an aggregate score of **68.9** on **SP-Bench** (Table 1) and **0.6811** on **Cell Annotation** (Table 3). On **CRC-CODEX** (Table 2), SP-Mind attains **0.9526 ± 0.0283** Pearson *r*, **0.9016 ± 0.1234** Spearman *ρ*, **0.7274 ± 0.1097** correlation-matrix cosine similarity, **0.3683 ± 0.0935** MMD, and **0.5134 ± 0.1053** spatial Spearman correlation.
>
> Here is the link to the result table:
>
> https://i.postimg.cc/TYNdHKTB/table-1.png
>
>
> **W2 & Q2.** *“Beyond execution success, have the authors evaluated the biological validity of the generated outputs...?”*
>
> **A.** Yes. We would like to clarify that, in Appendix C.3, **Output Fidelity** is already defined as part of execution success, so SP-Bench does not treat a query as successful based on mere code completion / sequential tool call. In our evaluation, the **Output Fidelity** criterion was manually assessed by **three bioinformatics experts**, who verified that the generated outputs were biologically valid and contained sufficient information suitable to be passed to the next workflow stage. Combined with other criteria mentioned, this yields a rigorous evaluation protocol; we will make this evaluation protocol more explicit in the revised manuscript.
>
> **Q3.** *”Failure case analysis.”*
>
> **A.** Yes. A representative failure mode is **file-management / state-tracking error**: after successfully completing several stages, the agent may place an intermediate result into the wrong folder or directory structure. Frequency ~15%. A second failure mode arises in **background subtraction**, where it requires additional input engineering (e.g., adapting `markers.csv` into compatible format). SP-Mind can usually handle this when background subtraction is an isolated task, but in longer chained workflows it may fail to sufficiently inspect the error (frequency ~40%). We believe these failures mainly stem from the difficulty of **long-context reasoning, state management, and task decomposition** in extended workflows.

---

> > ### Author Rebuttal · Reviewer_igYA · 2026-04-03
> >
> > Thank you to the authors for the detailed and constructive rebuttal. I appreciate the clarifications and the additional analyses provided.
> > - The newly added repeated-run experiments help address my concern regarding run-to-run variance. While the number of runs remains limited, the results provide useful evidence that the system exhibits reasonable stability.
> > - The clarification on output fidelity and the involvement of domain experts are also helpful. This partially addresses my concern regarding biological validity. However, I still believe the evaluation would be further strengthened by incorporating more explicit quantitative or standardized metrics to complement expert assessment.
> > - The added failure case analysis is valuable and improves the transparency of the system’s limitations, particularly in highlighting issues related to long-context reasoning and state management.
> >
> > Regarding the concern on technical contribution, I acknowledge the authors’ clarification that the contribution lies in the integrated system design, domain-specific skill templates, and the introduction of SP-Bench. I agree that these components are non-trivial and practically meaningful for spatial proteomics workflows. That said, my original assessment still holds that the core methodology largely builds upon existing agent paradigms (e.g., ReAct-style reasoning and tool-based execution). As such, the contribution is best characterized as system-level integration and domain adaptation rather than fundamentally new methodological advances.
> >
> > Additionally, the related work section would benefit from a more comprehensive positioning within the rapidly evolving literature on agents in biological application. In particular, several works provide a broader perspective on agent architectures, reasoning paradigms, and biomedical applications e.g.,
> > - Gao, S., Fang, A., Huang, Y., Giunchiglia, V., Noori, A., Schwarz, J. R., ... & Zitnik, M. (2024). Empowering biomedical discovery with AI agents. Cell, 187(22), 6125-6151.
> > - Qi, C., Wang, W., Jiang, S., Liu, Q., Song, X., Fang, H., & Wei, Z. (2026). Artificial Intelligence agents for biological research: a survey. Briefings in Bioinformatics, 27(1), bbag075.
> > Including and discussing these works would help better contextualize the contribution of this paper within the current landscape.
> >
> > Overall, I find the work to be technically solid, well-motivated, and potentially impactful for domain-specific scientific agents, particularly due to the benchmark and system design. In light of the authors’ clarifications and additional analyses, I am willing to increase my overall score.

---

> > > ### Author Response · Authors · 2026-04-03
> > >
> > > We sincerely appreciate your acknowledgement that the added repeated-run analysis, expert-based output fidelity assessment, and failure-case discussion have improved the clarity and transparency of the paper. We are also deeply encouraged by your recognition that the integrated system design, domain-specific skill templates, and SP-Bench together constitute a meaningful contribution to autonomous spatial proteomics analysis.
> > >
> > > We agree that the paper would benefit from broader positioning within the fast-moving literature on AI agents for biological research, and we will expand the related work section accordingly. Thank you!

---

### Decision · Program_Chairs · 2026-04-30

**Decision:**

Accept (regular)

**Comment:**

This paper presents SP-Mind, an LLM-based autonomous agent for spatial proteomics analysis, along with SP-Bench, the first standardized benchmark for this domain (102 tasks, 18 categories). All four reviewers acknowledge the work as well-motivated, technically solid, and empirically strong. SP-Mind consistently outperforms existing biomedical agent baselines on both the proposed benchmark and downstream biological tasks. Initial concerns regarding technical novelty, evaluation rigor, and lack of ablation studies were thoroughly addressed in the authors' rebuttal, which provided new experiments including multi-run variance analysis, skill-template ablations (showing a 6.5-point drop on SP-Bench without templates), and detailed failure case characterization. All reviewers confirmed their concerns were resolved, with three raising their scores. The benchmark and system offer significant, actionable value to the computational biology community.